# DNA damage in embryonic neural stem cell determines FTLDs' fate via early-stage neuronal necrosis

Hidenori Homma[1,*], Hikari Tanaka[1,*], Meihua Jin[1], Xiaocen Jin[1], Yong Huang[1], Yuki Yoshioka[1], Christian JF Bertens[1,2,3], Kohei Tsumaki[2], Kanoh Kondo[1], Hiroki Shiwaku[1,4], Kazuhiko Tagawa[1], Hiroyasu Akatsu[5], Naoki Atsuta[6], Masahisa Katsuno[6], Katsutoshi Furukawa[7], Aiko Ishiki[8], Masaaki Waragai[9], Gaku Ohtomo[10], Atsushi Iwata[10], Takanori Yokota[11], Haruhisa Inoue[12,13], Hiroyuki Arai[8], Gen Sobue[6], Masaki Sone[1,2], Kyota Fujita[1,*], Hitoshi Okazawa[1,14]

The early-stage pathologies of frontotemporal lobal degeneration (FTLD) remain largely unknown. In VCP[T262A]-KI mice carrying VCP gene mutation linked to FTLD, insufficient DNA damage repair in neural stem/progenitor cells (NSCs) activated DNA-PK and CDK1 that disabled MCM3 essential for the G1/S cell cycle transition. Abnormal neural exit produced neurons carrying over unrepaired DNA damage and induced early-stage transcriptional repression-induced atypical cell death (TRIAD) necrosis accompanied by the specific markers pSer46-MARCKS and YAP. In utero gene therapy expressing normal VCP or non-phosphorylated mutant MCM3 rescued DNA damage, neuronal necrosis, cognitive function, and TDP43 aggregation in adult neurons of VCP[T262A]-KI mice, whereas similar therapy in adulthood was less effective. The similar early-stage neuronal necrosis was detected in PGRN[R504X]-KI, CHMP2B[Q165X]-KI, and TDP[N267S]-KI mice, and blocked by embryonic treatment with AAV–non-phospho-MCM3. Moreover, YAP-dependent necrosis occurred in neurons of human FTLD patients, and consistently pSer46-MARCKS was increased in cerebrospinal fluid (CSF) and serum of these patients. Collectively, developmental stress followed by early-stage neuronal necrosis is a potential target for therapeutics and one of the earliest general biomarkers for FTLD.

## Introduction

Frontotemporal lobar degeneration (FTLD) is the third largest group of neurodegenerative dementia following Alzheimer's disease (AD) and Lewy body dementia. FTLD has been classified into several groups based on the types of aggregated proteins. The largest group is FTLD with aggregation of TDP43 (transactive response [TAR] DNA-binding protein of 43 kD), called FTLD-TDP. Inclusion body myopathy associated with Paget disease of bone and fronto-temporal dementia with FTLD-TDP (Watts et al, 2004; Schröder et al, 2005) is caused by valosin containing protein (VCP) mutations. In particular, the T262A mutation is linked to typical motor and speech symptoms along with inclusion body myopathy (Spina et al, 2013).

VCP (also called TERA or p97) is a protein of the ATPase associated with various activities (AAA+) superfamily, whose members are involved in diverse cellular functions including membrane transfer from the ER to the Golgi apparatus (Ye et al, 2001), ubiquitin–proteasome–dependent protein degradation (Dai & Li, 2001), reassembly of the Golgi apparatus after mitosis (Rabouille et al, 1998), ER-associated protein degradation (ERAD) (Rabinovich et al, 2002), ribosome-associated protein degradation (Brandman et al, 2012; Defenouillère et al, 2013; Verma et al, 2013; Brandman & Hegde, 2016), membrane fusion (Uchiyama et al, 2002; Uchiyama & Kondo, 2005), suppression of apoptosis via DIAP (Rumpf et al, 2011), and autophagy (Ju et al, 2009). VCP contributes to DNA double-strand break (DSB) repair by non-homologous end joining (NHEJ) (Acs et al, 2011; Meerang et al, 2011; Fujita et al, 2013; van den Boom et al, 2016) in non-proliferating cells. Furthermore, recent work showed that VCP regulates homologous recombination (Meerang et al, 2011; Bergink et al, 2013; van den Boom et al, 2016; Torrecilla et al, 2017), another DSB repair mechanism active in proliferating cells.

Other genes causally related to FTLD, such as progranulin (PGRN), charged multivesicular body protein 2b (CHMP2B), and TAR

[1]Department of Neuropathology, Medical Research Institute, Tokyo Medical and Dental University, Tokyo, Japan  [2]Department of Biomolecular Science, Faculty of Science, Toho University, Chiba, Japan  [3]School for Mental Health and Neuroscience (MHeNs), University Eye Clinic Maastricht, Maastricht University, Maastricht, The Netherlands  [4]Department of Psychiatry, Graduate School of Medical and Dental Sciences, Tokyo Medical and Dental University, Tokyo, Japan  [5]Department of Community-Based Medical Education, Nagoya City University Graduate School of Medical Sciences, Nagoya, Japan  [6]Department of Neurology, Nagoya University Graduate School of Medicine, Nagoya, Japan  [7]Division of Community Medicine, Tohoku Medical and Pharmaceutical University, Sendai, Japan  [8]Department of Geriatrics and Gerontology, Division of Brain Science, Institute of Development, Aging and Cancer, Tohoku University, Sendai, Japan  [9]Department of Neurology, Higashi Matsudo Municipal Hospital, Chiba, Japan  [10]Department of Neurology, The University of Tokyo, Graduate School of Medicine, Tokyo, Japan  [11]Department of Neurology, Graduate School of Medical and Dental Sciences, Tokyo Medical and Dental University, Tokyo, Japan  [12]Center for iPS Cell Research and Application (CiRA), Kyoto University, Kyoto, Japan  [13]Drug-Discovery Cellular Basis Development Team, RIKEN BioResource Center, Kyoto, Japan  [14]Center for Brain Integration Research, Tokyo Medical and Dental University, Tokyo, Japan

Correspondence: okazawa-tky@umin.ac.jp
*Hidenori Homma, Hikari Tanaka, and Kyota Fujita contributed equally to this work

DNA-binding protein of 43 kD (TDP43), are also involved in DNA damage repair through up-regulation of genes involved in this process (Bandey et al, 2014), repair of nuclear envelope rupture (Raab et al, 2016; Willan et al, 2019), and participation in NHEJ mediated by the Ku70/Ku80 complex (Mitra et al, 2019), respectively.

Impairment of DNA damage repair in neurodegeneration has been suggested by multiple studies with a number of molecules. In Huntington's disease, DNA damage repair proteins such as HMGB1, Ku70, and VCP were identified as interacting proteins of mutant huntingtin that are impaired by the abnormal interaction (Qi et al, 2007; Enokido et al, 2010; Fujita et al, 2013). In spinocerebllar ataxia, multiple DNA damage repair proteins such as HMGB1, VCP, and RpA1 were shown functionally impaired (Qi et al, 2007; Fujita et al, 2013; Ito et al, 2015a; Taniguchi et al, 2016). Recently, DNA damage repair genes were shown to be genetic modifiers of multiple polyglutamine diseases (Jones et al, 2017; Ross & Truant, 2017) and the impairment of DNA damage repair is now recognized as a common pathology. In FTLD pathology, Ku80 and some other DNA damage response (DDR) genes were activated to trigger apoptotic signal cascade (Lopez-Gonzalez et al, 2019), suggesting that pre- and post-DNA damage molecular events are involved in the pathologies of neurodegenerative diseases in general.

In this study, we generated a knock-in mouse model (VCP$^{T262A}$-KI mice) of FTLD carrying the VCP gene mutation. These mice exhibited developmental microcephaly, which was recovered in adulthood, as well as structural, cellular, and molecular abnormalities that we characterized. Comprehensive phosphoproteome analysis revealed that in NSCs, impairment of the DNA repair function of VCP$^{T262A}$ caused accumulation of DNA damage, activation of kinases that mediate DDR signaling, abnormal phosphorylation of minichromosome maintenance complex component 3 (MCM3), and cell cycle arrest. The embryonic pathology persisted in the adult brain as DNA damage, TRIAD necrosis (Hoshino et al, 2006; Mao et al, 2016a; Tanaka et al, 2020) and TDP43 aggregation. Gene therapy at the embryonic stage, but not in adulthood, restored normal adult phenotypes, demonstrating the significance of DNA repair in embryonic brains long before the symptomatic onset of VCP-linked FTLD (FTLD-VCP).

Moreover, we showed that DNA damage stress in the embryonic stage was common to three other FTLD-linked KI mice carrying mutations in PGRN, CHMP2B, or TDP43. Moreover, based on transcriptional repression coupled with DNA damage, we demonstrated that DNA damage carried over from NSCs to differentiated neurons caused early-stage necrosis. We confirmed the presence of similar necrosis in postmortem brains of human FTLD patients, as well as elevated levels of pSer46-MARCKS, a TRIAD marker, in CSF and serum of living patients. Comprehensive proteome analysis of the four models suggested that necrosis led to TDP43 protein aggregation pathology via PKCγ activation.

# Results

## Generation of a new mouse model of FTLD-VCP

We generated a targeting vector to introduce the T262A mutation into the mouse *Vcp* gene (Fig S1A), transfected linearized targeting vector into ES cells, and ultimately obtained seven mutant ES lines in the C57BL/6J (BL/6) background (Fig S1B). The sequences surrounding the *Vcp* gene mutation are highly conserved between mouse and human (Fig S1C), supporting the idea that the point mutation in the mouse gene reflects the corresponding human mutation. Three of seven mutant embryonic stem (ES) lines were microinjected into blastocysts and returned to the uterus. Only one line yielded chimeric mice (F0), from which we generated two lines of heterozygous VCP$^{T262A/+}$ mice (VCP$^{T262A}$-KI mice). The two lines were maintained by crossing male VCP$^{T262A}$-KI mice with female C57BL/6J mice. Mice were used for analysis after six generations of crossing. The phenotypes such as external appearance and body weight gain of the two lines did not differ significantly (Fig S1D).

VCP$^{T262A}$-KI mice developed cytoplasmic TDP43 aggregates in neurons of frontal associated cortex (FrA) and other regions of the cerebral cortex at 6 mo of age (Fig S2A); the TDP43 in the aggregates was ubiquitinated (Fig S2B). Nuclear p62 aggregates (Fig S2C) and cytoplasmic fused in sarcoma aggregates were detected at low frequencies (Fig S2D), as shown in the summary (Fig S2E). No abnormal staining for TDP43 or ubiquitin was detected in embryonic cerebral cortex at embryonic day (E)13 or E15 (Fig S2F).

Behavioral analyses revealed abnormal anxiety-related behaviors from 6 mo of age (Fig S3A), impaired spatial memory in the Morris water maze test from 6 mo of age (Fig S3A), and abnormal fear-related memories in the elevated plus maze test and in fear-conditioning test from 12 mo of age (Fig S3A). Moreover, dendritic spines were less abundant, and their rate of elimination in the cerebral cortex was elevated (Fig S3B and C), before the onset of memory impairment and cognitive decline. Together, these findings indicated that the VCP$^{T262A}$-KI mouse is a suitable model for FTLD-VCP.

## Developmental microcephaly of FTLD-VCP mouse model

At E18, brain weight was smaller in VCP$^{T262A}$-KI mice than in background controls, and the cerebral cortex was thinner (Fig 1A). The brain was also smaller from 1 to 3 mo after birth, although this difference was diminished after 6 mo of age (Fig 1B), and cortex thickness became equivalent to that of background mice at 6 mo (Fig 1C). Western blotting revealed that levels of total VCP proteins (wild type VCP + VCP$^{T262A}$) were similar between VCP$^{T262A}$-KI and background mice during the course of the experiments (Fig 1D). Although no macroscopic structural abnormalities were detected in adult brain at 6 mo of age (Fig 1C), immunohistochemistry with layer-specific markers revealed different patterns (Fig 1E) in the mutant mice: Cux1-positive and FOXP1-positive layers (layers II–III and III–V) were thinner in VCP$^{T262A}$-KI mice, whereas the TBR1-positive layer (layer VI) was thicker (Fig 1F), indicating that microscopic structural abnormalities remained in adulthood brains of VCP$^{T262A}$-KI mice.

Golgi staining of VCP$^{T262A}$-KI mice at 6 mo of age revealed that deep cortex layers were disproportionally thickened, whereas superficial layers became thinner (Fig 1G). The size of neurons became larger (Fig 1H) and the numbers of dendrites were increased in the external pyramidal layer of VCP$^{T262A}$-KI mice (Fig 1I).

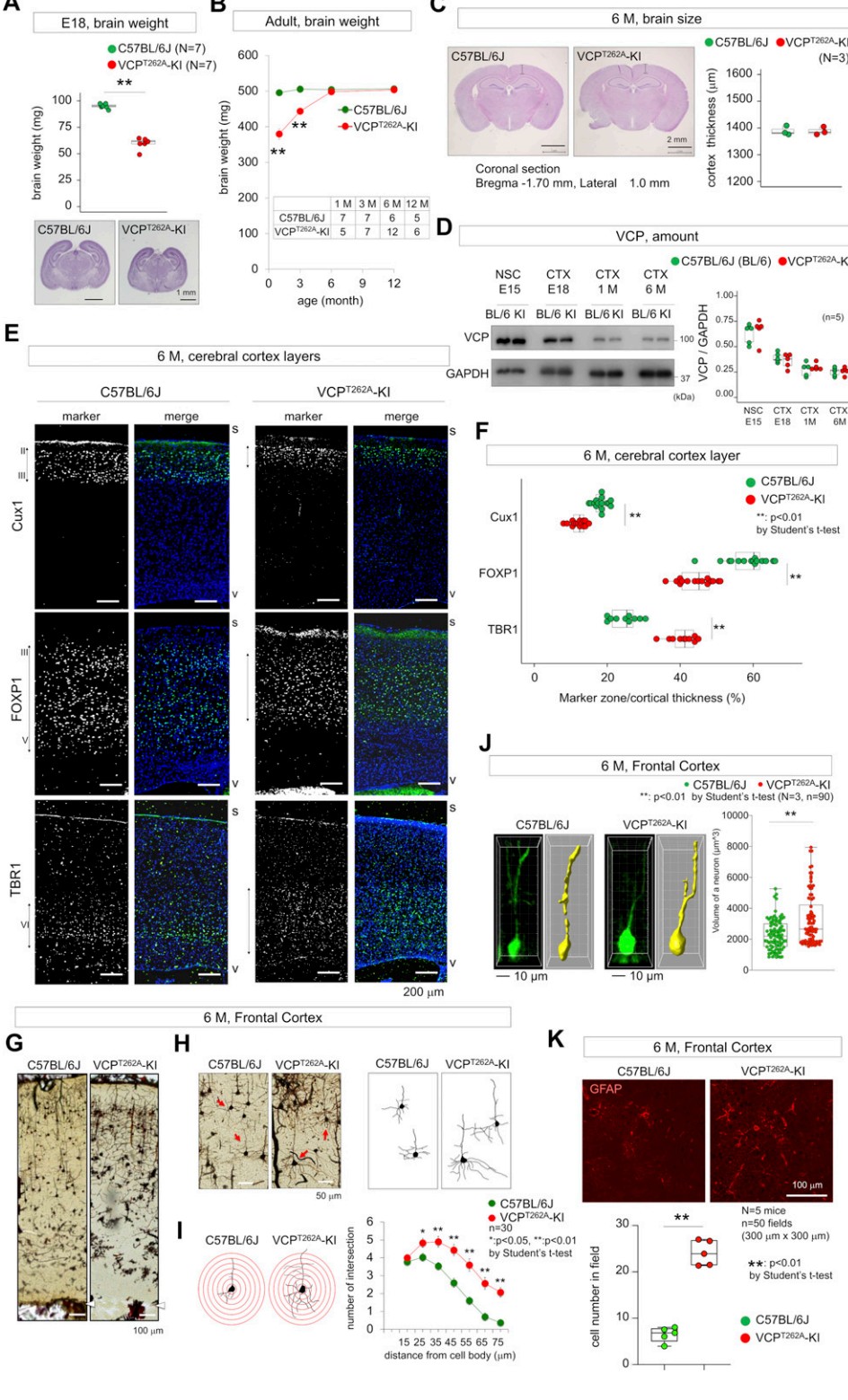

**Figure 1. Developmental and adult phenotypes of VCP$^{T262A}$-KI mice.**

**(A)** Brain weights of heterozygous VCP$^{T262A}$-KI and littermate C57BL/6J mouse embryos were measured at E18. Lower panels show coronal sections of the brains. No architectural change was detected in the brains of VCP$^{T262A}$-KI embryos, but the cerebral volume and cortical thickness were obviously reduced. Asterisks indicates $P < 0.01$ in $t$ test. **(B)** Brain weights of VCP$^{T262A}$-KI mice were measured at 1, 3, 6, and 12 mo of age. Numbers of samples are shown in the table. Asterisks indicates $P < 0.01$ in $t$ test. **(C)** Coronal sections of brains. Brain size and architecture were similar between VCP$^{T262A}$-KI and background C57BL/6J mice at 6 mo. Tissue sections were made from three mice (N = 3) at Bregma –1.7 mm. 10 sections (technical replicates) per sample were made. Cortex thickness was measured at 1.0 mm lateral from midline. **(D)** VCP expression levels in neural stem cells (E15) and total cerebral cortex at E18, 1 and 6 mo did not differ between VCP-KI mice and sibling non-transgenic mice (C57BL/6J). **(E)** Immunohistochemistry with layer markers. Cux1 for layer II and III, FOXP1 for layer III–V, and Tbr1 for layer VI. Cerebral cortex outer surface (s) and ventricular surface (v) are indicated in each panel. **(F)** Quantitative analysis of layer thickness. Asterisk indicates $P < 0.01$ in $t$ test. N (mouse number) = 3, and 10 sections were made per sample. **(G)** Golgi staining of cortex revealed disproportional cortical layers of VCP$^{T262A}$-KI mice. Arrowheads indicate ventricular surface. **(H)** High magnification of neurons in the external pyramidal layer. Tracing of neurons in the external pyramidal layer (arrow). **(I)** Dendrite arborization was measured by the number of dendrite crossing at the indicated distances from the center of the cell body. Double asterisks indicate $P < 0.01$ in $t$ test. N (mouse number) = 3; n (neuron number) = 30. **(J)** A single neuron volume was calculated by Surface of Imaris (ver. 7.7.2) and compared between 90 neurons of C57BL/6J and VCP$^{T262A}$-KI mice. **(K)** Cerebral cortex tissues were stained with anti-GFAP antibody to quantify glia density. Box plots show medians, quartiles, and whiskers, which represent data outside the 25$^{th}$–75$^{th}$ percentile range. Source data are available for this figure.

Imaris-based analysis of images obtained by two-photon microscopy also indicated increase in single-neuron volumes (Fig 1J). In addition, density of glia in the external pyramidal layer of frontal cortex was also increased in VCP$^{T262A}$-KI mice (Fig 1K). These factors could explain adulthood recapture of brain size in VCP$^{T262A}$-KI mice.

## DNA damage in embryonic NSCs and adult neurons of FTLD-VCP model

In U2OS cells, the VCP$^{T262A}$ mutation resulted in slower accumulation of VCP-EGFP fusion protein to DNA damage foci induced by micro-irradiation than normal VCP (Fig 2A), leading to slower DNA damage repair. Consistent with this, the level of DNA damage was elevated in NSCs and differentiated neurons in embryonic cerebral cortex at E13 and E15, as revealed by immunohistochemistry (Fig 2B) and Western blotting (Fig 2C). Primary NSCs from E15 embryonic forebrains of VCP$^{T262A}$-KI mice also reproduced the accumulation of DNA damage (Fig 2D). Accumulation of DNA damage was sustained in adult cerebral cortex until 3 and 6 mo of age (Fig 2E), whereas aggregates of phospho-TDP43 were observed after 6 mo of age (Fig 2F). Such DNA damage accumulation was also observed in immunohistochemistry and Western blot analysis of postmortem brains from human FTLD patients (Fig 2G and H). In such a terminal stage of human FTLD, DNA damage was significantly correlated with TDP43 aggregation in neurons (Fig 2G, right table).

## DNA damage in embryonic NSCs causes microcephaly via G1/S arrest

We next addressed the mechanism underlying developmental microcephaly. First, we investigated the change of the M phase that could cause microcephaly (Woods et al, 2005; Thornton & Woods, 2009). M-phase cells (i.e., pH3-positive cells) were not altered in abundance at the ventricular/subventricular zone (VZ/SVZ) in VCP$^{T262A}$-KI mouse embryos at E13 or E15 (Fig S4A). Moreover, N-cadherin staining revealed no change in the equal or unequal division of cell membranes from apical neural stem/progenitor cells (Fig S4B). Likewise, the mitotic plane angle, revealed by anti-γ-tubulin antibody, was not altered (Fig S4C).

However, NSCs prepared from E15 embryonic forebrains of VCP$^{T262A}$-KI mice exhibited a delay in proliferation (Fig S5A). FACS analysis of VCP$^{T262A}$-KI NSCs revealed G1/S arrest instead of G2/M arrest (Fig S5B). Moreover, in vivo analysis of cell cycle phase of NSCs by cumulative labeling of NSCs revealed elongation of G1 phase and total cell cycle time (Tc) (Fig S5C). Consistently with the result of pH3-positive cells, the duration of M-phase time was not changed in VCP$^{T262A}$-KI NSCs (Fig S5D). Consistently with the result of FACS analysis of VCP$^{T262A}$-KI NSCs, G1 time (G1) and S-phase time (Ts), deduced from the graph, were elongated (Fig S5E).

Co-staining for BrdU and the pan–cell cycle marker Ki67 revealed an increase in the proportion of BrdU-labeled cells that exited the cell cycle (Fig S6A), indicating that the increase in the G1 fraction promoted neuronal differentiation from NSCs. Such excessive neurogenesis theoretically depletes NSC pools in the VZ, causing a type of developmental microcephaly (Jayaraman et al, 2018). Actually, immunohistochemistry of embryonic cortex with BrdU and MAP2, a mature neuron marker, confirmed excessive neurogenesis at E15 (Fig S6B), and immunohistochemistry of apical and basal neural stem cell markers, Sox2 and Tbr2, confirmed depletion of the NSC pool in VZ/SVZ of VCP$^{T262A}$-KI mice at E10 and E15 (Fig S6C).

The frequency of cell death in NSCs and differentiating neurons were not changed when it was examined by in vivo annexin-V staining (Schrevens et al, 1998) (Fig S6D) instead of by TUNEL staining (nick-end labeling) that might also detect DNA damage in living cells as false-positive apoptosis.

## DNA damage–responsive phosphorylation of MCM3 arrests NSCs at G1/S

To elucidate the molecular mechanisms of partial arrest of the G1/S transition in VCP$^{T262A}$ NSCs, we performed comprehensive phosphoproteome analysis of NSCs, which revealed changes in the phosphorylation of regulatory proteins involved in the G1/S transition (Fig S7A). For example, nestin, a marker of neural stem/progenitor cells that is essential for self-renewal of NSCs (Park et al, 2010), was highly phosphorylated at multiple sites (Fig S7A). However, the candidate most relevant to the G1/S transition was MCM3 (minichromosome maintenance helicase 3), a DNA replication licensing factor or pre-replication complex component involved in initiating DNA replication and the G1/S transition (Remus et al, 2009; Li et al, 2015). We also detected phosphorylation of minichromosome maintenance complex component 2 (MCM2), which has a similar function in G1/S transition. However, phosphorylation of MCM3 at Thr719 was increased > ninefold, much more strongly than phosphorylation of MCM2 at Ser41 (Fig S7A). Western blots confirmed a marked increase in phosphorylation of MCM3, but not MCM2 (Fig 3A). Functional categorization of proteins detected by phosphoproteomics, based on the Kyoto encyclopedia of genes and genomes (KEGG) pathway database (https://www.genome.jp/kegg/pathway.html), indicated that phosphorylation was concentrated on proteins involved in DNA replication such as MCM3, MCM2, and Lig1 (Fig S7B), supporting the significance of the G1/S transition in NSC pathology. Molecular network analysis of proteins whose phosphorylation was altered in NSCs (Fig S7A) provided further support for the roles of MCM3 as network hubs (Fig 3B). Among 71 proteins whose phosphorylation was changed in VCP$^{T262A}$ mutant NSCs, 34 proteins were included in two-edge networks from MCM3, MCM2, and Lig1, respectively (Fig 3B).

To address the upstream mechanism leading to MCM3 phosphorylation, we used NetworKIN (http://networkin.info/index.shtml) and found that multiple DDR kinases (Blasina et al, 1999; Bartek & Lukas, 2001; Potapova et al, 2009; Satyanarayana & Kaldis, 2009; Enserink & Kolodner, 2010) might phosphorylate MCM3 at Thr719 (Fig 3C). In vitro phosphorylation followed by mass spectrometry revealed that CDK1 and DNA-PK could actually phosphorylate MCM3 at Thr719 (Fig 3D), whereas other DDR kinases such as ataxia telangiectasia mutated (ATM), CDK2, CDK4, and DAPK1 could not (Fig 3D). DNA-PK had strong kinase activity at this residue, although NetworKIN did not predict it (Fig 3D). Western blotting of DDR kinases, including DNA-PK and ATM, confirmed that CDK1, CDK2, DNA-PK, ATM, and Chk2 were activated in VZ+SVZ tissues prepared from cerebral cortex of E15 embryos (Fig 3E), whereas CDK4, DAPK1, and Chk1 were not (Fig 3E). Immunohistochemistry confirmed that CDK1 and DNA-PK were in NSCs of VCP$^{T262A}$-KI mouse embryos at E15 (Fig 3F). The active form of DNA-PK (pDNA-PK) was abundant in apical NSCs in which the level of DNA damage was elevated, as demonstrated by the presence of γH2AX/53BP1 foci and phosphorylation of MCM3 at Thr719 (Fig 3F), whereas the active form of CDK1 (pCDK1) was predominantly observed in basal NSCs (Fig 3F). Moreover, siRNA-mediated knockdown of CDK1 and DNA-PK suppressed the increase in pThr719-MCM3 in primary VCP$^{T262A}$ mutant NSCs, confirming that the two kinases were

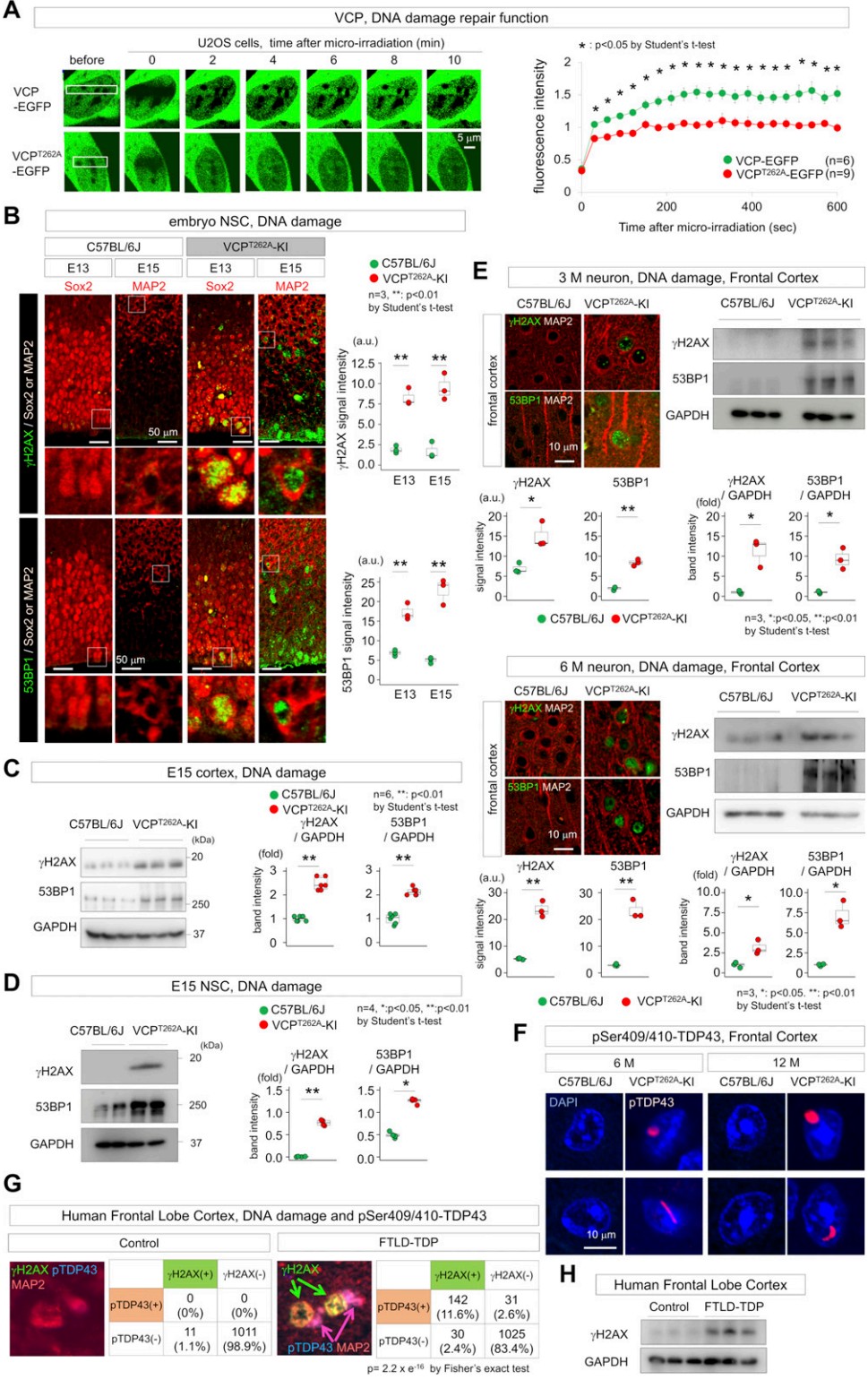

**Figure 2. DNA damage from NSCs to adult neurons of VCP$^{T262A}$-KI mice.**
**(A)** Time-lapse imaging of VCP-EGFP assembly in linear irradiated areas containing double-strand breaks. After transfection with VCP-EGFP or VCP$^{T262A}$-EGFP, the nuclei of U2OS cells were micro-irradiated, and assemblies of VCP-EGFP and VCP$^{T262A}$-EGFP were observed. Right graph shows quantitative analysis of the recovery of VCP-EGFP signals in the linear bleached areas. At each time-point after micro-irradiation, EGFP signal intensities were quantified and normalized to the area signal before micro-irradiation (white box). Means and s.e.m. are indicated. Asterisks indicate statistical differences ($P < 0.01$, $t$ test). **(B)** Immunohistochemistry with anti-γH2AX or anti-53BP1 antibodies in cerebral cortex (VZ/SVZ) at E13 and E15. Right graph shows percentage of VZ/SVZ cells with γH2AX or 53BP1 foci. **(C)** Western blot analyses of γH2AX and 53BP1 in cortex tissue at E15. **(D)** Western blot analyses of γH2AX and 53BP1 in NSC spheres prepared from embryonic brains of heterozygous VCP$^{T262A}$-KI and C57BL/6J mice at E15. Right graphs show quantitative analysis of band intensities. **(E)** Immunohistochemistry and Western blot of double-strand break markers γH2AX and 53BP1 reveal elevated DNA damage in adult cerebral neurons of VCP$^{T262A}$-KI mice at 3 and 6 mo. **(F)** Immunohistochemistry of phosphorylated TDP43 in cerebral cortex neurons of VCP$^{T262A}$-KI mice at 6 and 12 mo. **(G)** Immunohistochemistry of γH2AX and phosphorylated TDP43 in postmortem brains of human FTLD patients. γH2AX was barely detectable in control brains (non-neurological disease patient, N = 6). Representative images of double-positive neurons and γH2AX-positive/pTDP43-negative neurons from FTLD patients (N = 5) are shown. Tables show classification of staining patterns in human cortical neurons in frontal lobe of three controls and three FTLD patients. Staining patterns of γH2AX and pTDP43 in MAP2-positive neurons were positively related (Fisher's exact test, $P = 2.2 \times 10^{-16}$). **(H)** Western blot analysis of γH2AX in postmortem brains of human FTLD patients and controls. Box plots show medians, quartiles, and whiskers, which represent data outside the 25th–75th percentile range. Source data are available for this figure.

responsible for MCM3 phosphorylation at Thr719 (Fig 3G). A similar investigation determined that CDK1 and DNA-PK were the kinases responsible for phosphorylation of MCM2, a homologous protein involved in regulating the G1/S transition, at Ser41 (Fig S7C and D).

## Non-phosphorylated MCM3 rescues FTLD-VCP in vitro and in vivo

To confirm the functional impairment of pThr719-MCM3, we attempted to rescue the G1/S transition in VCP$^{T262A}$ mutant NSCs

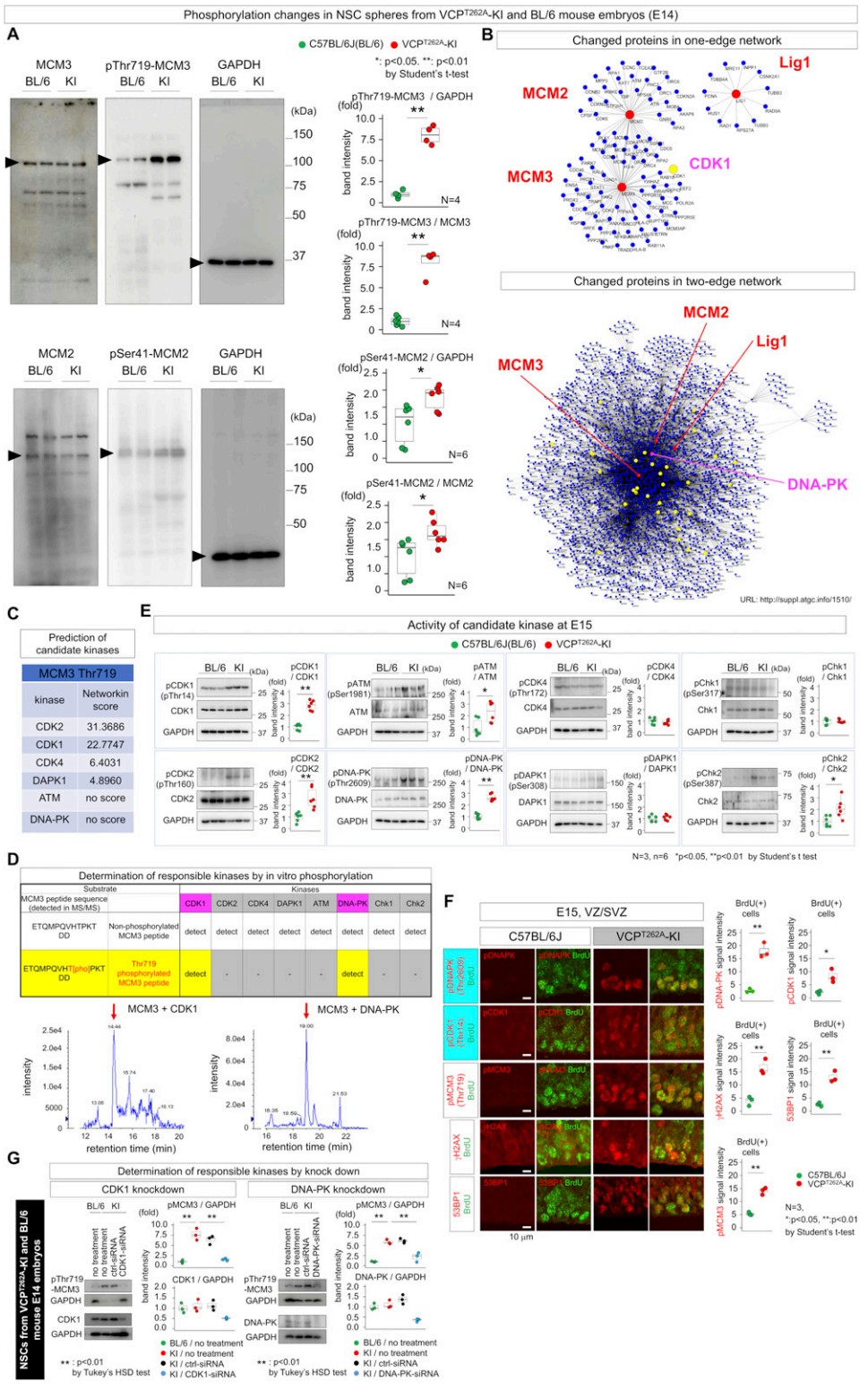

**Figure 3. MCM3 phosphorylation in NSCs as a trigger of FTLD phenotypes.**

**(A)** Western blots of phosphorylated MCM3 and MCM2 with NSCs prepared from VCP[T262A]-KI mouse embryos at E14. Right graphs show quantitative analyses. **(B)** Molecular network of proteins connected to MCM3, MCM2, and Lig1 ("DNA duplication" group in Fig S7B). One-edge and two-edge networks were generated using the PPI database (see the Materials and Methods section). Red nodes indicate MCM3, MCM2, and Lig1. Yellow nodes indicated proteins that changed in the phosphoproteome analysis (Fig S7A). **(C)** Candidate kinases for MCM3 phosphorylation at Thr719 predicted by NetworKIN. **(D)** Mass analysis of in vitro phosphorylation of MCM3 peptide by candidate kinases. "Detected" in upper tables indicates detection of phosphorylated or non-phosphorylated peptide fragments of MCM3 at Thr719 by mass spectroscopy at >95% confidence. Lower images show raw data. **(E)** Western blots revealed distinct levels of various DNA damage response kinases in developmental cortex at E15. Right graphs show quantitative analyses of three C57BL/6J and three VCP[T262A]-KI mice. **(F)** Immunohistochemistry of the active form of DNA-PK, the active form of CDK1, phosphorylated MCM3, and DNA damage markers (γH2AX and 53BP1). The active form of DNA-PK, phospho-MCM3 and DNA damage markers are detected in apical NSCs, whereas the active forms of CDK1 and DNA damage markers were detected in basal NSCs at E15. The right graph shows average signal intensities of 30 cells per field from each mouse (N = 3). **(G)** Effect of siRNA-mediated knockdown of each kinase on pThr719-MCM3 in primary NSCs prepared from VCP[T262A]-KI mouse E14 embryos. Box plots show medians, quartiles, and whiskers, which represent data outside the 25th–75th percentile range. Source data are available for this figure.

from KI mouse embryos. Wild-type MCM3 or a non-phosphorylated mutant (MCM3-T719A), but not phospho-mimetic mutants (MCM3-T719D and MCM3-T719E), restored the proliferation of VCP[T262A] mutant NSCs (Fig S8A) and G1/G0 accumulation (Fig S8B). Moreover,

similar experiments with human iPS cells (iPSCs) carrying the FTLD-linked L198W mutation confirmed that wild-type MCM3 and MCM3-T719A, but not MCM3-T719D and MCM3-T719E, rescued proliferation (Fig S8C) or the G1/S transition (Fig S8D).

Adeno-associated virus (AAV) vectors expressing normal VCP or non-phosphorylated mutant MCM3 (MCM3$^{T719A}$) were injected into pregnant mice carrying VCP$^{T262A}$-KI embryos at E10, and brains were dissected at E15, 1 and 6 mo of age (Fig 4A). At E15, brain weight (Fig S9A) and DNA damage accumulation (Fig S9B) were already normalized. Brain weight remained normal at 1 mo of age, indicating that AAV-VCP-FLAG or AAV-MCM3$^{T719A}$-FLAG injected at E10 rescued the developmental microcephaly of VCP$^{T262A}$-KI mice (Fig 4B). The result was further confirmed by the normalization of brain size (Fig 4C) and cortical layers (Fig 4D) at 6 mo. Immunohistochemistry (Fig 4E) and Western blot analysis (Fig 4F) at 6 mo revealed recovery from DNA damage in cortical neurons in animals that received AAV injection at E10. Consistent with this, the Morris water maze and Y-maze tests revealed that injection of AAV-VCP-FLAG or AAV-MCM3$^{T719A}$-FLAG at E10 restored spatial memory of VCP$^{T262A}$-KI mice at 6 mo (Fig 4G).

In contrast to gene therapy at the embryonic stage, adult gene therapy had only a partial rescue effect. In these experiments, we injected AAV-pCMV-VCP-FLAG or AAV-pCMV-MCM3$^{T719A}$-FLAG, in which the CMV promoter drives expression of target genes in both NSCs and neurons (Chung et al, 2002), into the subarachnoid space surface of the cerebral cortex of VCP$^{T262A}$-KI mice at 5 mo of age (Fig 4H). AAV-pCMV-VCP-FLAG, but not AAV-MCM3$^{T719A}$-FLAG recovered DNA damage (Fig 4I) and spatial memory in VCP$^{T262A}$-KI mice at 6 mo (Fig 4J). AAV-pCMV-VCP-FLAG and AAV-pCMV-MCM3$^{T719A}$-FLAG influenced adult neurogenesis and corrected excessive neuronal differentiation from adult NSCs (Fig S9C), although this effect was limited because of the low rate of neuron production. The abnormal layer structure was not cured by post-development gene therapy (Fig S9D).

Intriguingly, TDP43 aggregation pathology of VCP$^{T262A}$-KI mice at 6 mo of age was altered by AAV-VCP-FLAG or AAV-MCM3$^{T719A}$-FLAG administered in utero, but not by them injected at 5 mo of age in adulthood (Fig 4K). The result indicated that phosphorylation and aggregation of TDP43 in adult neurons are downstream of impairment of DNA damage in NSCs due to VCP mutation. Although it is clear that embryonic VCP pathology in NSCs definitely contributed to TDP43 pathology in neurons of the cerebral cortex in adulthood, further investigation is needed to fill the gap between the two pathologies and reveal what kinds of mechanisms induce, modify, or prevent the TDP43 aggregation-dependent gain or loss of function.

### DNA damage in neurons causes TRIAD necrosis in FTLD-VCP

It is well known that DNA damage in cells substantially affects general transcription levels including yes-associated protein (YAP) gene (Maldonado et al, 1996; Hoshino et al, 2006; Ju et al, 2006) and in differentiated neurons causes TRIAD necrosis relevant to YAP expression level (Hoshino et al, 2006; Mao et al, 2016a; Tanaka et al, 2020). Immunohistochemistry of cerebral cortex of VCP$^{T262A}$-KI and their background (C57BL/6J) mice at 1 mo of age revealed that YAP levels were substantially reduced in neurons harboring DNA damage, as reflected by nick-end-labeling, and surrounded by pSer46-MARCKS (Fig 5A). Given that the pattern matched well with morphological feature of TRIAD necrosis (Hoshino et al, 2006; Mao et al, 2016a; Tanaka et al, 2020), we examined the frequency of TRIAD necrosis in the four FTLD mouse models using the specific marker pSer46-MARCKS (Fujita et al, 2016; Tanaka et al, 2020). As expected,

TRIAD necrosis (weak DAPI signal and high pSer46-MARCKS signal) was observed from 1 to 12 mo of age, and the peak of frequency was at 3 mo of age (Fig 5B). Western blotting showed the increase in the level of pSer46-MARCKS in total cortex of VCP$^{T262A}$-KI mice (Fig 5C) and electron microscopy revealed remarkable ER expansion in cortical neurons (Fig 5D), further supporting the presence of TRIAD necrosis in FTLD pathology.

Because TRIAD necrosis shares some morphological features with other forms of nonapoptotic cell death like pyroptosis (Brennan & Cookson, 2000; Fink & Cookson, 2006; Labbé & Saleh, 2008; Bergsbaken et al, 2009) and paraptosis (Sperandio et al, 2000), and because YAP was recently implicated in pyroptosis (Huang et al, 2020), we tested the relationship of their specific molecular markers, caspase 1 and caspase 9, to TRIAD-specific markers YAP (Hoshino et al, 2006; Mao et al, 2016a; Yamanishi et al, 2017), KDEL (Mao et al, 2016a; Yamanishi et al, 2017), and pSer46-MARCKS (Tagawa et al, 2015; Fujita et al, 2016) using brain tissues of VCP$^{T262A}$-KI mice and human patients of sporadic FTLD. Cortical neurons in ischemic lesions, used as a positive control, exhibited various types of necrosis stained by the pyroptosis marker caspase-1, the paraptosis marker caspase-9, and the necroptosis markers pRIP3 and pMLKL (Fig 5E). However, TRIAD neurons with low YAP and DAPI signals surrounded by pSer46-MARCKS signals were not stained by caspase-1, caspase-9, or pRIP3-pMLKL (Fig 5E), implying that TRIAD is distinct from pyroptosis, paraptosis or necroptosis.

### Generality of developmental stress and TRIAD necrosis in multiple FTLDs

We next asked whether the developmental stress associated with VCP-linked FTLD pathology could be generalized to other FTLDs caused by different genes. Previously, we generated PGRN$^{R504X}$-KI mice that reflect the human pathology (Fujita et al, 2018). In this study, we generated two additional FTLD mouse models harboring mutation of TDP43 (TDP$^{N267S}$-KI) or CHMP2B (CHMP2B$^{Q165X}$-KI) (Fig S10A–D), which reproduced human FTLD-TDP or FTLD-UPS pathology (Fig S11) and cognitive impairment (Fig S12). Comparison of the four models revealed similar decreases in brain weight at birth and recovery after that (Fig S13A). DNA damage detected by immunohistochemistry of γH2AX was carried over from E15 NSCs to adult cortical neurons via embryonic neurons in PGRN$^{R504X}$-KI, CHMP2B$^{Q165X}$-KI, and TDP$^{N267S}$-KI mice similarly to VCP$^{T262A}$-KI mice (Fig S13B).

YAP was substantially reduced and pSer46-MARCKS signal was elevated, in such neurons with DNA damage in nick-end-labeling of PGRN$^{R504X}$-KI, CHMP2B$^{Q165X}$-KI and TDP$^{N267S}$-KI mice at 1 mo of age (Fig 6A). Similarly to VCP$^{T262A}$-KI mice, TRIAD necrosis was observed from 1 to 12 mo of age in PGRN$^{R504X}$-KI, CHMP2B$^{Q165X}$-KI, and TDP$^{N267S}$-KI mice, their frequency reached a peak at 3 mo of age (Fig 6B). Western blotting confirmed the increase in pSer46-MARCKS in PGRN$^{R504X}$-KI mice (Fig 6C). Activation of DDR kinases such as CDK1 and DNA-PK, and phosphorylation of MCM3 at Thr719 were confirmed in all four mouse models by Western blotting with E15 NSCs (Fig S13C). E15 NSCs prepared from the four FTLD models proliferated slowly, and FACS analysis revealed their partial G1/S arrest (Fig S13D). NSCs differentiated from human PGRN$^{R493X}$-iPS cells also exhibited delayed proliferation and partial G1/S arrest (Fig S13E).

DNA damage broadly affects transcriptional gene expression and can trigger TRIAD necrosis due to suppression of the *Yap* gene

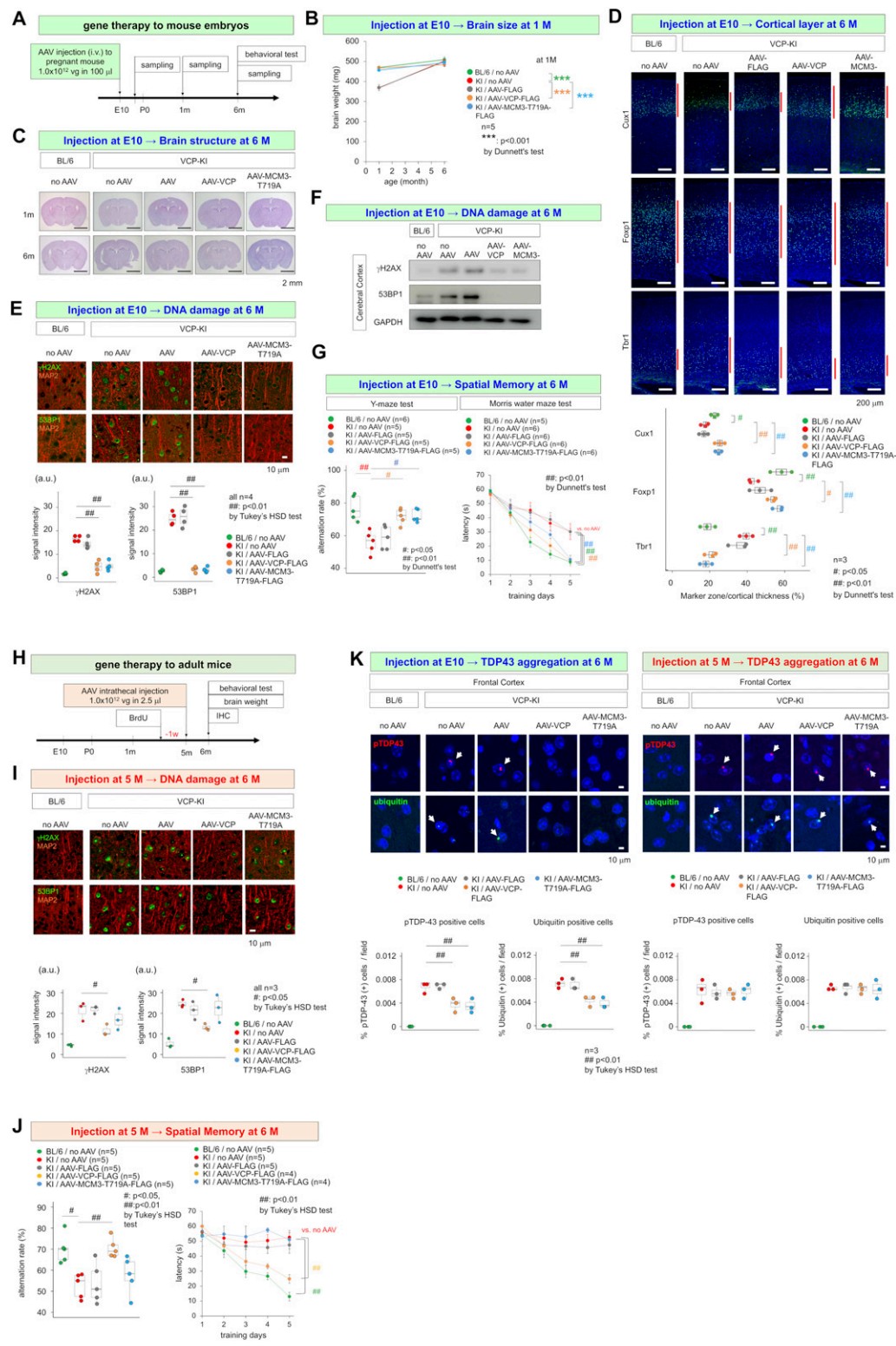

**Figure 4. Non-phosphorylated MCM3 rescues phenotypes of VCP^T262A-KI mice.**
**(A)** Experimental protocol for AAV injection into mouse embryos and subsequent examinations. AAV-pNestin-VCP-FLAG, AAV-pNestin-MCM3^T719A-FLAG, and AAV-pNestin-FLAG were injected into pregnant mice at E10 via the tail vein (1.0 × 10^12 vg in 100 μl). Newborn mice were reared by their mothers until weaning, and then their genotypes were examined. At the ages of 1 or 6 mo, brains were dissected, weighed, and subjected to immunohistochemistry. At the age of 6 mo, behavioral tests (Y-maze test and Morris water maze test) were performed. **(B)** Total brain weight of VCP^T262A-KI mice (male) treated at E10, from 1 to 6 mo of age. **(C)** Coronal sections of brains of VCP^T262A-KI mice at 6 mo. At 1 mo, AAV-pNestin-VCP-FLAG and AAV-pNestin-MCM3^T719A-FLAG, but not AAV-pNestin-FLAG, rescued brain size. **(D)** Abnormal cortical layer

(Hoshino et al, 2006; Mao et al, 2016a, 2016b; Yamanishi et al, 2017). Consistently, gene chip and Western blot analyses revealed reduction in *Yap* mRNA and YAP protein in cortical neurons from four FTLD mouse models at 1 mo of age (Fig S13F and G).

We further examined existence of TRIAD necrosis in human FTLD, and detected achromasia of neurons, reflecting abnormality of ER with ribosomes in postmortem brains of human FTLD patients (Figs 6D and S14). Immunohistochemistry revealed ER ballooning with anti-KDEL antibody, reactive phosphorylation of MARCKS with anti–pSer46-MARCKS antibody, and reduced level of nuclear YAP protein with anti-YAP antibody (Fig 6E), which reflected the feature of TRIAD necrosis. Electron microscopy also confirmed ER expansion in cortical neurons (Fig 6F). Interestingly, skein-like stains of pSer409/410-TDP43 (Lowe et al, 1988; Leigh et al, 1991; Robinson et al, 2013) were frequently co-positive for pSer46-MARCKS (Fig 6G, yellow arrow).

TRIAD necrosis induces ER expansion and intracellular $Ca^{2+}$ release. Quantitative mass analysis revealed activation of PKCγ (pThr514- and pThr655-PKCγ), a response to intracellular $Ca^{2+}$ leaked from the expanded ER, in all four FTLD models from 3 mo of age (Table S1). NetPhos 3.1 predicted that multiple phosphorylation sites of TDP43 are targets of PKC (http://www.cbs.dtu.dk/services/NetPhos/) (Fig S15). In vitro phosphorylation followed by Western blotting confirmed that PKCγ could phosphorylate TDP43 at Ser403/404 and Ser409/410 (Fig 6H), which are relevant to inclusion body formation (Hasegawa et al, 2008; Neumann et al, 2009). Levels of pSer403/404- and pSer409/410-TDP43 in cerebral cortex of three FTLD-TDP models were elevated from 6 mo of age, as revealed by Western blotting (Fig 6I).

Finally, we tested therapeutic effects of in utero administration of AAV-pCMV-MCM3[T719A]-FLAG, which cuts at the root of the common pathological pathway shared by multiple FTLD mouse models. The protocol was similar to that used for VCP[T262A]-KI mice (Fig 4A). As expected, in utero gene therapy rescued DNA damage (Fig 6J, upper panels), TRIAD necrosis (Fig 6J, middle panels), and TDP43 aggregation (Fig 6J, lower panels) in adult neurons of PGRN[R504X]-KI, CHMP2B[Q165X]-KI, and TDP[N267S]-KI mice at 6 mo of age. Consistent with this, the treatment also rescued the cognitive decline (Fig 6K) of PGRN[R504X]-KI, CHMP2B[Q165X]-KI, and TDP[N267S]-KI mice in adulthood, as in VCP[T262A]-KI mice.

Immunohistochemistry of caspase-1 revealed weak signals in a subset of cortical neurons of FTLD model mice, although the frequency of such cells was much lower than that of cells undergoing TRIAD (Fig S16). Caspase-9 signal was barely detectable in cortical neurons of FTLD model mice (Fig S16). Markers of necroptosis, such

as pRIP3 and pMLKL, were not co-detected in neurons of FTLD model mice at 1 or 3 mo of age (Fig S16), as in AD model mice and the brains of human AD patients (Tanaka et al, 2020). Immunohistochemistry of postmortem human brains with various necrosis and apoptosis markers confirmed the predominance of TRIAD in human sporadic FTLD (Fig S17).

Collectively, the data from four FTLD mouse models, mouse FTLD-models derived NSCs, human FTLD postmortem brains, and human FTLD-iPSC–derived NSCs supported the general concept across multiple FTLDs, which could possibly include sporadic FTLD, that developmental DNA damage stress contributes to adult pathology via TRIAD necrosis (Fig S18).

### CSF and serum pSer46-MARCKS as a biomarker of FTLD

Consistent with the findings, the level of pSer46-MARCKS, which is linked to TRIAD necrosis (Fujita et al, 2016; Tanaka et al, 2020), was elevated in CSF of human patients with FTLD (Fig 7A). In addition, CSF-pSer46-MARCKS was elevated in patients with amyotrophic lateral sclerosis (ALS), which is pathologically homologous to FTLD, as well as in AD patients, who also exhibit TRIAD necrosis (Tanaka et al, 2020) (Fig 7A). Receiver operating characteristic curve analysis indicated that CSF-pSer46-MARCKS could be a biomarker of FTLD (Fig 7B) and other neurodegenerative diseases associated with TRIAD necrosis (Fig 7B). Moreover, CSF-pSer46-MARCKS was correlated with CSF-neurofilament-L (CSF-NFL), but was more sensitive (Fig S19). Comparison of the serum of FTLD patients and normal controls suggested that serum-pSer46-MARCKS could be also used as a blood biomarker and confirmed the correlation with serum-neurofilament-L (Fig 7C).

## Discussion

The primary significance of this study is the discovery that developmental stress due to DDR triggered by accumulation of unrepaired DNA damage in NSCs contributes to initiation of the ultra-early stage FTLD pathology. In VCP[T262A]-KI mice, DNA repair in embryonic NSCs led to accumulation of DNA damage, DDR activation, MCM3 phosphorylation, partial G1/S arrest, excessive neurogenesis, depletion of the NSC pool, abnormal layer structure, and sustained DNA damage in neurons linked to early-stage neuronal necrosis (Fig S18). Impairment of DNA damage repair and resultant accumulation of DNA damage in neurons are now recognized as a common pathology across multiple neurodegenerative diseases (Qi et al, 2007; Enokido et al, 2010; Fujita et al, 2013; Jones et al, 2017;

structure was normalized by AAV-pNestin-VCP-FLAG and AAV-pNestin-MCM3[T719A]-FLAG. Cux1 (layers II–III), Foxp1 (layers III–V) and Tbr1 (Layers V–VI) were used as layer markers. **(E)** Immunohistochemistry of DNA damage markers in VCP[T262A]-KI mice treated at E10, at 6 mo of age. **(F)** Western blotting for DNA damage markers in VCP[T262A]-KI mice treated at E10 at 6 mo of age. **(G)** Morris water maze test and Y-maze test showed recovery of spatial memory impairment at 6 mo of age in VCP[T262A]-KI mice injected with AAV-pNestin-VCP-FLAG or AAV-pNestin-MCM3[T719A]-FLAG at E10. Graphs show latency to platform at day 5 of training in the Morris water maze test and alteration rate in the Y-maze test. #*P* < 0.05, ##*P* < 0.01 (Dunnett's test). **(H)** Protocol for gene therapy in VCP[T262A]-KI mice at 5 mo (adulthood). **(I)** Immunohistochemistry for DNA damage markers was performed in 6-mo-old VCP[T262A]-KI mice that received gene therapy at 5 mo. AAV-pCMV-VCP-FLAG, but not AAV-pCMV-MCM3[T719A]-FLAG, decreased DNA damage in cortical neurons. **(J)** Morris water maze test and Y-maze test showed recovery of spatial memory impairment at 6 mo of age in VCP[T262A]-KI mice injected with AAV-pCMV-VCP-FLAG or AAV-pCMV-MCM3[T719A]-FLAG at E10 at 5 mo of age. Graphs show latency to platform at day 5 of training in the Morris water maze test and alternation rate in the Y-maze test. #*P* < 0.05, ##*P* < 0.01 (Dunnett's test). **(K)** TDP43 aggregation in cortical neurons at 6 mo of VCP[T262A]-KI mice that were treated with AAV-pNestin-VCP-FLAG and AAV-pNestin-MCM3[T719A]-FLAG at E10 or with AAV-pCMV-VCP-FLAG and AAV-pNestin-MCM3[T719A]-FLAG at 5 mo. The number of cells with TDP43 aggregation decreased significantly in mice that received gene therapy at E10. Box plots show median, quartiles, and whiskers, which represent data outside the 25th–75th percentile range. Source data are available for this figure.

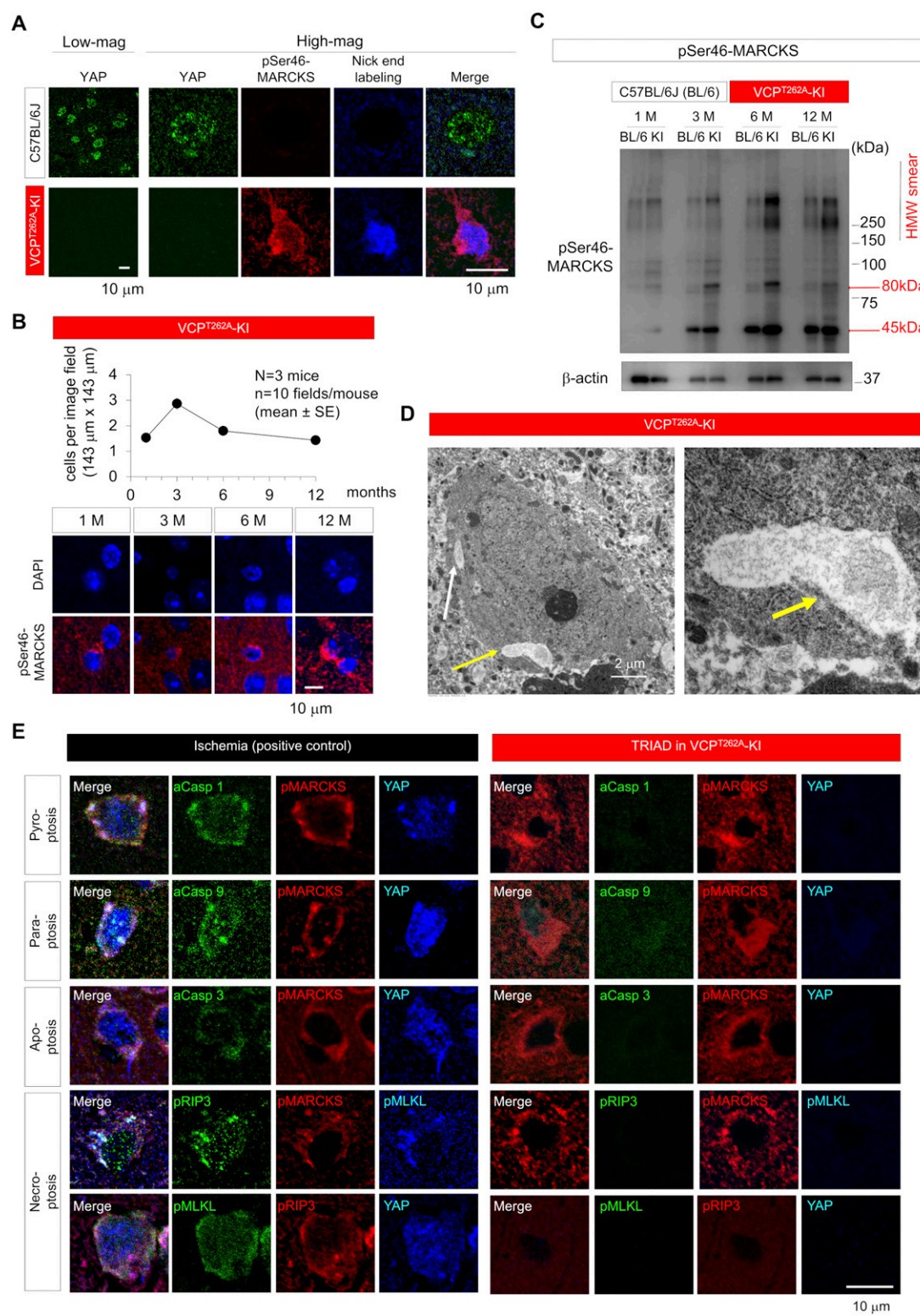

**Figure 5. Early-stage neuronal TRIAD necrosis in VCP^T262A^-KI mice.**
**(A)** Immunohistochemistry of YAP, pSer46-MARCKS, damaged DNA (nick-end-labeling) in cerebral cortex of VCP^T262A^-KI mice and their background (C57BL/6J) at 1 mo of age. **(B)** Chronological changes in the frequency of TRIAD necrosis in cerebral cortex of VCP^T262A^-KI mice. Necrotic neurons were counted in randomly selected visual fields (n = 10, N = 3, 143 × 143 μm). Representative images of necrotic neurons associated with pSer46-MARCKS are shown in the lower panels. **(C)** Western blots reveal elevated levels of pSer46-MARCKS in cerebral cortex during aging, which was more prominent in VCP^T262A^-KI mice. **(D)** Electron microscopy images confirmed ER expansion of necrotic neurons in cerebral cortex of VCP^T262A^-KI mice (white and yellow arrows). High-magnification images of vacuoles (yellow arrow) are shown. **(E)** Distinguishing TRIAD of VCP^T262A^-KI mice (1 mo) from paraptosis, pyroptosis, and necroptosis in the brains after ischemia.
Source data are available for this figure.

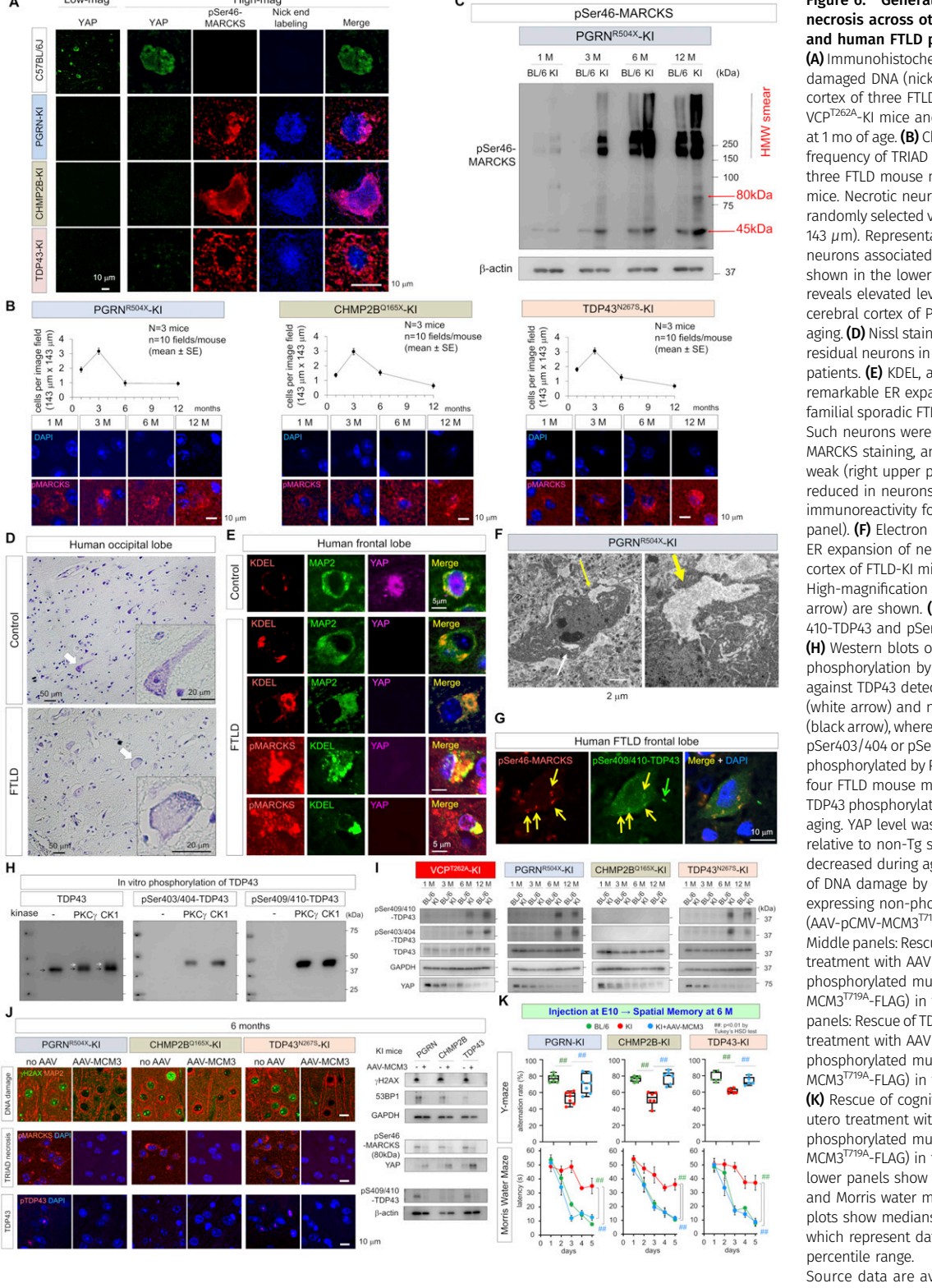

**Figure 6. Generality of neuronal TRIAD necrosis across other FTLD mouse models and human FTLD patients.**

**(A)** Immunohistochemistry of YAP, pSer46-MARCKS, damaged DNA (nick-end-labeling) in cerebral cortex of three FTLD mouse models other than VCP$^{T262A}$-KI mice and their background (C57BL/6J) at 1 mo of age. **(B)** Chronological changes in the frequency of TRIAD necrosis in cerebral cortex of three FTLD mouse models other than VCP$^{T262A}$-KI mice. Necrotic neurons were counted in randomly selected visual fields (n = 10, N = 3, 143 × 143 µm). Representative images of necrotic neurons associated with pSer46-MARCKS are shown in the lower panels. **(C)** Western blot reveals elevated levels of pSer46-MARCKS in cerebral cortex of PGRN$^{R504X}$-KI mice during aging. **(D)** Nissl staining revealed chromatolysis of residual neurons in occipital lobes of human FTLD patients. **(E)** KDEL, an ER marker, reveals remarkable ER expansion in neurons of non-familial sporadic FTLD patients (left upper panels). Such neurons were accompanied by pSer46-MARCKS staining, and DAPI staining became weak (right upper panels). YAP signals were reduced in neurons with ER expansion and immunoreactivity for pSer46-MARCKS (left lower panel). **(F)** Electron microscopy images confirmed ER expansion of necrotic neurons in cerebral cortex of FTLD-KI mice (white and yellow arrows). High-magnification images of vacuoles (yellow arrow) are shown. **(G)** Co-staining of pSer409/410-TDP43 and pSer46-MARCKS of a neuron. **(H)** Western blots of TDP43 protein after in vitro phosphorylation by PKCγ or CK1. Antibody against TDP43 detected both phosphorylated (white arrow) and non-phosphorylated TDP43 (black arrow), whereas specific antibody against pSer403/404 or pSer409/410 detected only TDP43 phosphorylated by PKCγ or CK1. **(I)** Western blots of four FTLD mouse models revealed changes in TDP43 phosphorylation and total YAP level during aging. YAP level was reduced in FTLD model mice relative to non-Tg sibling mice, and also decreased during aging. **(J)** Upper panels: Rescue of DNA damage by in utero treatment with AAV expressing non-phosphorylated mutant MCM3 (AAV-pCMV-MCM3$^{T719A}$-FLAG) in four FTLD models. Middle panels: Rescue of TRIAD necrosis by in utero treatment with AAV expressing non-phosphorylated mutant MCM3 (AAV-pCMV-MCM3$^{T719A}$-FLAG) in four FTLD models. Lower panels: Rescue of TDP43 aggregation by in utero treatment with AAV expressing non-phosphorylated mutant MCM3 (AAV-pCMV-MCM3$^{T719A}$-FLAG) in four FTLD models. **(K)** Rescue of cognitive impairment by in utero treatment with AAV expressing non-phosphorylated mutant MCM3 (AAV-pCMV-MCM3$^{T719A}$-FLAG) in four FTLD models. Upper and lower panels show the results of the Y-maze test and Morris water maze test, respectively. Box plots show medians, quartiles, and whiskers, which represent data outside the 25$^{th}$–75$^{th}$ percentile range.

Source data are available for this figure.

Ross & Truant, 2017; Lopez-Gonzalez et al, 2019). Interestingly, Ku70 and Ku80, whose heterodimer recruit DNA ligase to DNA DSBs for NHEJ in neurons, are implicated in the mechanism (Enokido et al, 2010; Lopez-Gonzalez et al, 2019). Loss of function of these repair molecules in the upstream and subsequent activation of DDR signaling also mediated by these repair molecules in the downstream of pathology might accelerate neurodegeneration in a pathological loop, which is consistent with accumulating knowledge that DNA

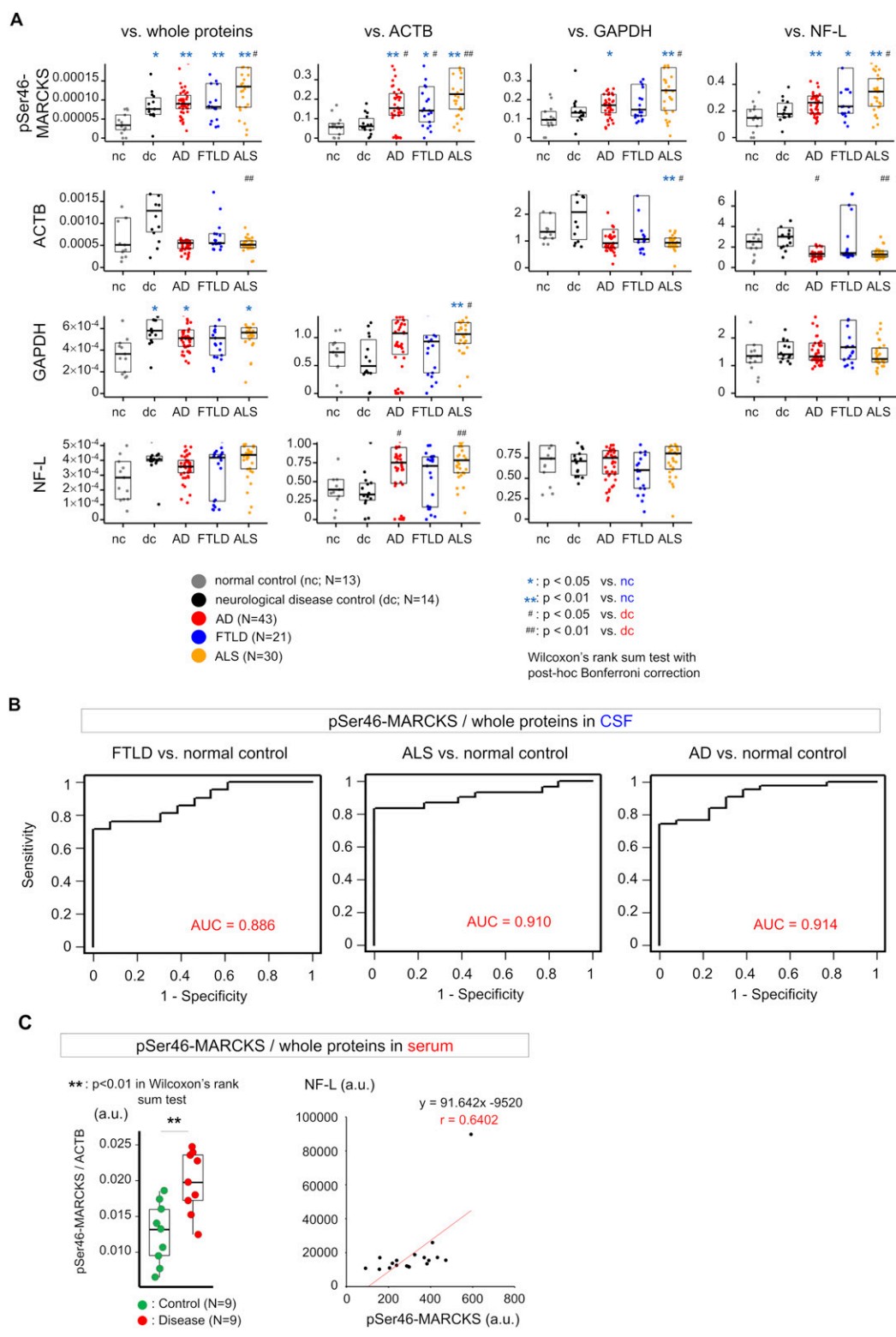

**Figure 7. Elevated levels of CSF- and serum-pSer46-MARCKS in FTLD patients.**
**(A)** Increase in the level of pSer46-MARCKS in CSF from human patients with sporadic FTLD, ALS, or Alzheimer's disease. The increase in the level of pSer46-MARCKS was greater than that of neurofilament L (NF-L). **(B)** Receiver operating characteristic analysis revealed the utility of CSF-pSer46-MARCKS for diagnosis of neurodegenerative diseases associated with TRIAD necrosis. AUC, area under the curve. **(C)** Mass spectrometry–based quantification revealed an elevated level of pSer46-MARCKS in serum from the disease group (seven ALS and two FTLD patients) (left graph) and a positive correlation between pSer46-MARCKS and neurofilament-L in serum (right graph). Box plots show medians, quartiles, and whiskers, which represent data outside the 25th–75th percentile range.
Source data are available for this figure.

damage repair molecules could both enhance and suppress neurodegeneration (Kovtun et al, 2007; Qi et al, 2007; Enokido et al, 2010; Barclay et al, 2014).

Our second major finding is the discovery that DNA damage drives early-stage neuronal necrosis in FTLD. Morphological and biochemical analyses revealed ER expansion, reduction in YAP levels in the nucleus, and immunoreactivity for pSer46-MARCKS, all of which are hallmarks of TRIAD necrosis (Hoshino et al, 2006; Mao et al, 2016a, 2016b; Yamanishi et al, 2017; Tanaka et al, 2020). TRIAD necrosis in FTLD did not associate with necroptosis markers such as pRIP1/3 and pMLKL, consistent with our previous results regarding TRIAD necrosis in human patients with mild cognitive impairment or AD (Tanaka et al, 2020), mouse models of AD (Tanaka et al, 2020), and mouse models of Huntington's disease (Mao et al, 2016a). Markers of pyroptosis, but not paraptosis or necroptosis, were detected in cortical neurons, but at a far lower frequency than in neurons positive for TRIAD markers, indicating that TRIAD is primarily responsible for early-stage neuronal necrosis.

The third major finding from this study is the discovery that multiple types of FTLD share a pathological cascade from impairment of DNA repair in NSCs to aggregation pathology in adulthood, mediated by early-stage neuronal necrosis. Impairment of DNA repair in FTLD increases production of differentiated neurons in which DNA damage is carried over from NSCs. DNA damage leads to ER instability in neurons, triggering an increase in intracellular $Ca^{2+}$ and activation of PKC, leading to phosphorylation and aggregation of TDP43. TRIAD necrosis could also trigger aggregation of TDP43 via the HMGB1–TLR4–PKC axis in surrounding neurons. It remains unclear why CHMP2B$^{Q165X}$-KI mice, a model of FTLD-UPS, generate p62 aggregates rather than TDP43 aggregates among four FTLD models. The question might be answered by a future study in which we perform comprehensive analyses of chronological changes.

The fourth unexpected finding from this study is developmental and recapturable microcephaly. The mechanism is revealed based on DNA damage in NSCs causing G1/S arrest, and the resultant increase in G0 fraction of NSCs leads to microcephaly, as reported in various human gene mutations responsible for microcephaly (Woods et al, 2005; Jayaraman et al, 2018). Therefore, microcephaly based on depletion of NSC pool and ultra-early stage pathology based on TRIAD necrosis of neurons are originated from the same root, DNA damage of NSCs (Fig S18). The unique feature of FTLDs is that their microcephaly is recapturable in adulthood. Our findings on the increase of neuronal volume and of glial number might explain the unique feature, although the underlying mechanism should be elucidated in details in the future. Various hypotheses would be possible. For instance, activation of Rb, which is known to regulate neuronal size (Lee et al, 1994), might be carried over from NSCs to neurons.

From the viewpoint of practical applications, identification of pSer46-MARCKS in CSF and serum as a biomarker and recovery of adult FTLD phenotype by gene therapy in utero might be extended in the future. Because most FTLD patients do not harbor any causative gene mutations, it is critical to determine whether the proposed scheme applies to non-familial FTLD. Notably in this regard, the presence of necrosis in postmortem brains and elevation of pSer46-MARCKS in CSF strongly supported the idea that similar pathological cascades occur in non-familial FTLD cases.

These findings, in turn, indicate that pSer46-MARCKS in CSF and serum could be a surrogate biomarker for ultra–early-phase pathology, and could thus be used to evaluate disease progression and therapeutic response.

On the other hand, gene therapy in utero needs to overcome multiple issues to address before developing it as a preventive and/or therapeutic method. First, exact prediction is currently impossible for a person without genetic mutation to develop FTLD in later life. Second, safety of gene therapy in utero has not been established, and comparison between risk and safety has not reached the stage of discussion. However, a certain surrogate biomarker, possibly including pSer46-MARCKS, could be discovered in the future that predict a sign of onset when a certain switching occurs from normal to abnormal life course at the very early time-point. If the safety of in utero gene therapy has established by technical improvements in the future, extremely large economical and human costs could be far beyond the risk of gene therapy in utero.

Collectively, our data reveal that developmental stress in NSCs and subsequent early-stage neuronal necrosis contribute to later stage adulthood pathologies across multiple familial FTLD pathologies, suggesting that the therapeutic time window for complete recovery of FTLDs may be much earlier than previously expected.

# Materials and Methods

### Generation of VCP$^{T262A}$-KI mice

We generated heterozygous knock-in mice carrying the VCPT262A mutation with the aid of Unitech (Fig S1). Knock-ins were constructed in the C57BL/6J background. For construction of the targeting vector, a 5.4-kb NotI–XhoI fragment amplified from a bacterial artificial chromosome (BAC) (ID: RP23-111G9 or RP23-124L1) was subcloned into PspOMI-XhoI of pBS-DTA (Unitech) and designated Plasmid 1. Similarly, a 2.9-kb BamHI-NotI fragment was amplified from the same BAC and subcloned into pBS-LNL(−), which contains a Neo cassette (loxP-Neo-loxP) (Unitech), yielding Plasmid 2. Plasmid 2 was digested with SalI and blunted, and then a 4.5-kb fragment (1.6-kb Neo cassette + 2.9-kb fragment) was cleaved out with NotI. A 0.8-kb XhoI–SmaI fragment with the T262A mutation was PCR-amplified from the same BAC and digested with XhoI and SmaI. The 0.8- and 4.5-kb fragments were subcloned into the XhoI–NotI sites of Plasmid 1, and the resultant plasmid was used as a targeting vector. After linearization of the targeting vector with NotI, the fragment was electroporated into ES cells of the C57BL/6J background. Genotyping of ES clones was performed by PCR using the following primers: 5′-CGTGCAATCCATCTTGTTCAAT-3′ (forward) and 5′-AAGACAGCTCCTCACTACCTAACAG-3′ (reverse), and positive clones were further confirmed by Southern blot analysis using 5′- and -3′-probes amplified from genomic DNA using primers 5′-TGATT-GAGTTGTAAAACCTTGTTCC-3′ and 5′-GATGACAAGCAGACTCCACATTAC-3′ (5′ probe) or 5′-ATGGAGATTGCTTTGATTTCAGG-3′ and 5′-ACCTGCAAA-GAGAAGATAGATTGAC-3′ (3′ probe). To confirm that the neomycin cassette was removed in F2 mice generated by crossing F1 and CAG-Cre

mice, a neomycin probe prepared by genomic PCR using primers 5′-GAACAAGATGGATTGCACGCAGGTTCTCCG-3′ and 5′-GTAGCCAACGCTATG-TCCTGATAG-3′ was used for Southern blot analysis.

VCP[T262A]-KI mice were discriminated from non-transgenic littermates by PCR using primers 5′-ATATGCTCTCACTGTATGGTATTGC-3′ and 5′-TCCAGATGAGCTTAGAAGATTAGAA-3′, which amplify the DNA fragment containing the LoxP sequence. PCR conditions were as follows: 35 cycles of 94°C for 10 s for denaturation, 56°C for 40 s for annealing, and 72°C for 40 s for extension. A 168-bp fragment and a 266-bp fragment were amplified from background and heterozygous VCP[T262A]-KI mice, respectively. Behavioral experiments were performed by multiple examiners. Genotyping was also performed by multiple examiners without exchange of information before behavioral tests. The results from each examiner were mixed for statistical tests, with power analysis to validate the sample size.

### Hematoxylin–Eosin staining

Mouse brains were fixed with 4% paraformaldehyde for 12 h, washed with PBS, and embedded in paraffin. Sagittal or coronal sections (5-$\mu$m thickness) were made using a microtome (Yamato Kohki Industrial Co., Ltd.). The sections were de-paraffinized by xylene, rehydrated, washed with water, and stained with Carrazzi's Hematoxylin solution (1.15938.0025; Merck) for 10 min at room temperature. After washing with tap water for 10 min, sections were stained with eosin solution for 5 min at room temperature (0.25% eosin diluted with 80% ethanol, 051-06515; Wako) and dehydrated with ethanol. Sections were immersed with xylene for clearing and covered. Images were obtained by light microscopy (BX53; Olympus).

### Golgi staining

For Golgi staining, VCP[T262A]-KI and C57BL/6 littermate mice were fixed by cardiovascular perfusion of fixation solution (4% paraformaldehyde and 20% glutaraldehyde in 0.05 M phosphate buffer). Brains were immersed in the same fixation solution for another 1 h at 4°C, washed with 0.1 M phosphate buffer, incubated with potassium dichromate solution (5% potassium dichromate, 4.875% chloral hydrate, 1.5% glutaraldehyde, and 2% PFA) for 7 d at room temperature, washed three times with ddH$_2$O, and incubated in 0.7% silver nitrate solution for 7 d at room temperature. Then, the brain samples were dehydrated with ethanol (70% for 1 h, 90% for 1 h, 100% for 2 h, and 50% ether plus 50% ethanol for 1 h), and embedded in celloidin solution (0.25% for 1 h, 5% for 12 h, and 10% for 48 h) at room temperature. 50-$\mu$m sections prepared with a vibrating microtome (Microm HM650V; Thermo Fisher Scientific) were dehydrated with 70% ethanol (three times), anhydrous alcohol, and 3:1 anhydrous alcohol:chloroform (three times); cleared with xylene (three times), and coverslipped with Marinol (#2009-3; Muto Pure Chemicals).

### Immunohistochemistry

5-$\mu$m sections were deparaffinized in xylene, re-hydrated, dipped in 0.01 M citrate buffer (pH 6.0), and microwaved at 120°C for 15 min. After permeabilization with 0.5% Triton X-100 containing PBS, sections were incubated with blocking solution (10% FBS containing

PBS) for 60 min at room temperature. Then, sections were incubated with primary antibody for 12 h at 4°C, washed with PBS three times at room temperature, and incubated with secondary antibodies at room temperature for 1 h. All the procedures were performed in parallel for the mouse groups being compared.

Antibodies used for immunohistochemistry were diluted as follows, rabbit anti-Cux1, 1:100 (sc-13024; Santa Cruz Biotechnology); rabbit anti-FOXP1, 1:200 (ab16645-100; Abcam); rabbit anti-Tbr1, 1:100 (ab31940; Abcam); goat anti-Sox2, 1:200 (sc-17320; Santa Cruz Biotechnology); chicken anti-Tbr2, 1:500 (AB15894; Millipore); mouse anti-GFAP Cy3-conjugated, 1:2,000 (C9205; Sigma-Aldrich), mouse anti-BrdU, 1:100 (347580; BD Bioscience); rabbit anti–phospho-histone H3, 1:500 (06-570; Millipore); rabbit anti-Ki67, 1:200 (NCL-L-Ki67-MM1; Novocastra Laboratories); mouse anti-γH2AX, 1:100 (Ser139, #05-636; Millipore); rabbit anti-53BP1, 1:5,000 (NB100-304; Novus Biologicals); rabbit anti-TDP43, 1:500 (G400; Cell Signaling Technology); rabbit anti–phospho-TDP43 (Ser409/410), 1:5,000 (TIP-PTD-P02; Cosmo Bio); mouse anti-ubiquitin, 1:1,000 (P4D1; Cell Signaling Technology); rabbit anti-fused in sarcoma, 1:200 (ab84078; Abcam); mouse anti-p62, 1:200 (#610497; BD Bioscience); mouse anti-pTau (AT-8), 1:1,000 (90206; Innogenetics); mouse anti-γ-tubulin, 1:1,000 (T5326; Sigma-Aldrich); mouse anti-cadherin, 1:500 (C1821; Sigma-Aldrich); mouse anti-MAP2, 1:50 (sc32791; Santa Cruz Biotechnology); rabbit anti-phospho-DNA-PK (pThr2609), 1:200 (600-401-494; Rockland Immunochemicals); rabbit anti-phospho-CDK1 (pThr14), 1:200 (ab58509; Abcam); rabbit anti-phospho-MCM3 (pThr719), 1:200 (TA313106; OriGene); rabbit anti-YAP, 1:50 (sc-15407; Santa Cruz Biotechnology); rabbit anti–phospho-MARCKS (Ser46), 1:2000 (GL Biochem [Shanghai] Ltd.); mouse anti-KDEL, 1:200 (ADI-SPA-827; Enzo Life Science); rabbit anti-MAP2, 1:1,000 (ab32454; Abcam); rabbit anti–cleaved caspase-1, 1:50 (#89332; Cell Signaling Technology); rabbit anti–cleaved caspase-9, 1:20 (#9509; Cell Signaling Technology); rabbit anti–cleaved caspase-9, 1:200 (#9661; Cell Signaling Technology); rabbit anti–pSer232-RIP3, 1:200 (ab195117; Abcam); rabbit anti-pSer345-MLKL, 1:400 (ab196436; Abcam); donkey anti-rabbit IgG, Cy3-conjugated, 1:500 (711-165-152; Jackson Laboratory); donkey anti-mouse IgG Alexa Fluor 488, 1:1,000 (A21202; Molecular Probes); donkey anti-mouse IgG, Cy3-conjugated, 1:500 (715-165-150; Jackson Laboratory); donkey anti-rabbit IgG Alexa Fluor 488, 1:1,000 (A21206; Molecular Probes); goat anti-chicken IgY, FITC-conjugated, 1:1,000 (103-095-155; Jackson ImmunoResearch Laboratories).

For multi-labeling, antibodies were labeled with Zenon Secondary Detection–Based Antibody Labeling Kits as follows: anti–cleaved caspase-1, anti–cleaved caspase-9, anti–cleaved caspase-3, anti–pSer232-RIP3, and anti–pSer345-MLKL (Zenon Alexa Fluor 647 Rabbit IgG Labeling Kit, Z-25308; Thermo Fisher Scientific); anti–pSer46-MARCKS (Zenon Alexa Fluor 488 Rabbit IgG Labeling Kit, Z-25302; Thermo Fisher Scientific); and anti–pSer345-MLKL, and anti–pSer46-MARCKS (Zenon Alexa Fluor 568 Rabbit IgG Labeling Kit, Z-25306; Thermo Fisher Scientific).

For nick-end-labeling, deparaffinized sections were washed three times at room temperature with PBS containing 0.1% Tween-20 (PBST). The washed sections were incubated with labeling reaction mix (Biotin-16-dUTP [11093070910; Roche] and Terminal Transferase [33335566001; Roche]) at 37°C for 2 h, washed with PBST three times at room temperature, and then incubated with Alexa

Fluor 594–conjugated streptavidin (S11227; Thermo Fisher Scientific) at room temperature for 1 h.

TDP43 or ubiquitin was enzymatically detected using the VEC-TASTAIN Elite ABC Standard Kit (PK-6100; Vector Laboratories) and the DAB Peroxidase Substrate Kit (SK-4100; Vector Laboratories). For DAB staining, sections were counterstained for 10 min at RT with Carrazzi's hematoxylin solution (1.15938.0025; Merck), washed with tap water for 10 min, and dehydrated with ethanol. Sections were cover-slipped after clearing with xylene.

## Nissl staining

Paraffin-embedded sections of human brain were deparaffinized and rehydrated. After washing with distilled water, sections were immersed for 15 min at 37°C in cresyl violet solution (300 ml of 0.1% cresyl violet in distilled water with five drops of 10% acetic acid solution, filtered before use). Sections were dehydrated with ethanol, cleared with xylene, and cover-slipped. Images were acquired under a light microscope (BX53; Olympus).

## Western blotting

Mouse cortical tissues (E15 and adult brain) were dissolved for 1 h at 4°C in lysis buffer (100 mM Tris–HCl [pH 7.5], 2% SDS) with protease inhibitor cocktail (#539134, 1:100 dilution; Calbiochem). After centrifugation (12,000$g$ × 10 min), the supernatants were added to an equal volume of sample buffer (62.5 mM Tris–HCl, pH 6.8, 2% [wt/vol] SDS, 2.5% [vol/vol] 2-mercaptoethanol, 5% [vol/vol] glycerol, and 0.0025% [wt/vol] bromophenol blue). The BCA method (Pierce BCA Protein Assay Kit; Thermo Fisher Scientific) was used to determine protein concentrations. Samples were separated by SDS–PAGE, transferred onto Immobilon-P membrane (Millipore) by a semi-dry method, blocked with 5% milk in TBST (10 mM Tris–HCl, pH 8.0, 150 mM NaCl, and 0.05% Tween-20), and reacted with primary and secondary antibodies diluted in TBST with 0.1% skim milk or Can Get Signal solution (Toyobo) as follows: mouse anti-γH2AX, 1:3,000 (Ser139, #05-636; Millipore); rabbit anti-53BP1, 1:30,000 (NB100-304; Novus Biologicals); rabbit anti–phospho-DNA-PK (pThr2609), 1:5,000 (600-401-494; Rockland Immunochemicals); rabbit anti–DNA-PK, 1:3,000 (sc-9051; Santa Cruz Biotechnology); rabbit anti–phospho-ChK1 (pSer317), 1:3,000 (#2344S; Cell Signaling Technology); mouse anti-ChK1, 1:3,000 (sc-8408; Santa Cruz Biotechnology); rabbit anti–phospho-ChK2, 1:3,000 (#2688S; Cell Signaling Technology): rabbit anti-ChK2, 1:1,000 (sc-9064; Santa Cruz Biotechnology); mouse anti–phospho-ATM (pSer1981), 1:3,000 (200-301-400; Rockland Immunochemicals); rabbit anti-ATM, 1:1,000 (PS85-100UG; Oncogene Research Products); rabbit anti–phospho-MCM2 (pSer41), 1:3,000 (A300-788A; Bethyl Laboratories Laboratory); goat anti-MCM2, 1:1,000 (sc-9839; Santa Cruz Biotechnology); rabbit anti–phospho-MCM3 (pThr719), 1:3,000 (TA313106; OriGene); rabbit anti-MCM3, 1:1,000 (#4012S; Cell Signaling Technology); rabbit anti–phospho-CDK1 (pThr14), 1:3,000 (ab58509; Abcam); mouse anti-CDK1, 1:3,000 (ab18; Abcam); rabbit anti-phospho CDK2 (pThr160), 1:1,000 (2561S; Cell Signaling Technology); rabbit anti-CDK2, 1:1,000 (ab7954; Abcam); rabbit anti-phospho CDK4 (pThr172), 1:1,000 (ab137675; Abcam); rabbit anti-CDK4, 1:1,000 (PA5-27827; Thermo Fisher Scientific); rat anti-HA, 1:2000 (11867423001; Sigma-Aldrich); mouse anti–phospho DAPK1 (pSer308), 1:1,000 (D4941; Sigma-Aldrich); rabbit anti-DAPK1, 1:1,000 (3008S; Cell Signaling Technology); mouse anti-tubulin, 1:3,000

(T8660; Sigma-Aldrich); mouse anti-VCP, 1:1,000 (612182; BD Bioscience); anti-GAPDH, 1:10,000 (MAB374; Millipore); rabbit anti-YAP, 1:1,000 (#14074; Cell Signaling Technology); rabbit anti–phospho TDP43 (Ser409/410), 1:6,000 (TIP-PTD-P02; Cosmo Bio); rabbit anti–phospho TDP43 (Ser403/404), 1:3,000 (TIP-PTD-P05; Cosmo Bio); rabbit anti-TDP43, 1:1,000 (ab109535; Abcam); HRP-linked anti-rabbit IgG, 1:3,000 (NA934; GE Healthcare); HRP-linked anti-mouse IgG, 1:3,000 (NA931; GE Healthcare). Incubation with primary antibodies was performed for 12 h at 4°C, and incubation with secondaries was performed for 1 h at room temperature. Proteins were detected using ECL Prime Western Blotting Detection Reagent (RPN2232; GE Healthcare) on an Image-Quant luminescence image analyzer.

## Neural stem precursor cell (NSC) preparation

NSCs were isolated from VCP$^{T262A}$-KI and littermate C57BL/6 mice embryos (E14) as described previously (Ito et al, 2015a; Mao et al, 2016a, 2016b). Embryos were washed with ice-cold PBS, and cerebral cortex was dissected with scalpel blades in a dish containing PBS. Cerebral cortex was cut into 1-mm cubes and dissociated into cells by repeated pipetting using a 10-ml plastic pipette (2-5237-04; AS ONE). After centrifugation (114$g$ × 5 min), supernatants were removed, and the cell pellet was resuspended with DMEM/F12 medium (12400-024; Gibco) containing 2% B27 supplement (17504-044; Gibco), 20 ng/ml basic FGF (G5071; Promega) and 20 ng/ml EGF (G5021; Promega), passed through a 70-$\mu$m cell strainer (352350; BD Falcon), and seeded onto 10-cm dishes. 7 d after preparation, NSCs were passaged and used for experiments.

## Generation and characterization of human VCP$^{L198W}$ cells

Patient dermal fibroblasts (GM20926) carrying the VCP$^{L198W}$ mutation, isolated from a FTLD-VCP patient, were obtained from Coriell Institute. iPS cells (iPSCs) with the *VCP* mutation were generated from the patient fibroblasts using episomal vectors for OCT3/4, Sox2, Klf4, L-Myc, Lin28, and p53-shRNA, as previously reported (Okita et al, 2013). Generated iPSCs were cultured on an SNL feeder layer with human iPSC medium (primate embryonic stem cell medium; ReproCELL) supplemented with 4 ng/ml basic FGF (Wako Chemicals) and penicillin/streptomycin.

For in vitro three-germ layer assays, iPSCs were dissociated and transferred to suspension plates with DMEM/Ham's F12 (DMEM/F12; Sigma-Aldrich) containing 20% knockout serum replacement (KSR; Life Technologies), 2 mM L-glutamine, 0.1 mM nonessential amino acids (NEAA; Invitrogen), 0.1 mM 2-mercaptoethanol (2-ME; Life Technologies), and 0.5% penicillin and streptomycin, and cultured to generate embryoid bodies. On day 8, embryoid bodies were moved onto gelatin-coated plates and differentiated for an additional 8 d, followed by immunostaining analysis. We confirmed the pluripotency of the resultant VCP$^{L198W}$-iPSCs.

## NSC differentiation from human iPSCs

VCP$^{L198W}$-iPSCs and control iPSCs (201B7; RIKEN BRC CELL BANK) were cultured on an SNL feeder cell layer in iPSC medium: Primate ES Cell medium (RCHEMD001; ReproCELL) supplemented with 4 ng/ml bFGF (100-18B; Peprotech) and penicillin/streptomycin (15140-122;

Gibco) at 37°C in a 5% $CO_2$ incubator. The iPSC medium was changed once daily. To generate neural stem cells, iPSC colonies were dissociated using Dissociation Solution for human ES/iPS Cells (RCHETP002; ReproCELL), and then the small clumps of cells were plated on an SNL feeder cell layer with iPSC medium containing 10 $\mu$M Y27632 (253-00513; Wako). Starting the next day, the medium was replaced with iPSC medium supplemented with 10 mM SB431542 (13031; Cayman Chemical), 10 mM CHIR99021 (13122; Cayman Chemical), and 5 mM Dorsomorphin (044-33751; Wako); this medium was replaced once a day for 8 d. iPS colonies were dissociated to single cells using Dissociation Solution for human ES/iPS Cells (RCHETP002; ReproCELL), plated on cell-repellent surface dishes (664970; Greiner Bio-One), and then cultured for 10 d with keratinocyte basal medium Neural Stem Cell medium (16050100; KOHJIN BIO) containing B27 (17504044; Thermo Fisher Scientific), 20 ng/ml bFGF (100-18B; Peprotech), 10 ng/ml hLIF (07690-31; Nacalai Tesque), 10 $\mu$M Y27632 (253-00513; Wako), 3 $\mu$M CHIR99021 (13122; Cayman Chemical), 2 $\mu$M SB431542 (13031; Cayman Chemical), and penicillin/streptomycin (Gibco) at 37°C in a 5% $CO_2$/4% $O_2$ (low $O_2$) incubator.

## Cell proliferation assays

Transfected or non-transfected NSCs were collected at the indicated time-points. Viable cell numbers were determined by trypan blue exclusion. Growth curves were indicated as the average ± SEM of viable cell number from four separate experiments. Statistical analyses were performed by ANOVA, $t$ test, or Tukey's HSD test as indicated.

## FACS scanning

Second-passage neurospheres (48 h after plating) from $VCP^{T262A}$-KI mice and littermate C57BL/6 mice were used for FACS. Cell were harvested, fixed with 70% ethanol in PBS for 24 h at −20°C, washed, and labeled with 40 $\mu$g/ml propidium iodide (P4170; Sigma-Aldrich) for 30 min at 37°C. For each analysis, nearly 10,000 cells were collected on a FACSCalibur system and analyzed with CELLQuest (Becton Dickinson).

## Cumulative labeling

Cumulative labeling for evaluating the cell-cycle phase was performed as described previously (Konno et al, 2008; Ito et al, 2015b) by injecting peritoneally into pregnant mice at E14. Cumulative labeling was performed by repeated peritoneal injection of BrdU (100 mg/kg of body weight, B5002; Sigma-Aldrich) at 3-h intervals into pregnant mice, which were euthanized 1, 1.5, 2, 3.5, 6.5, 9.5, 12, 15, and 24 h after the first BrdU injection. Embryonic brains were fixed with 4% paraformaldehyde and embedded with paraffin. Brain sections were deparaffinized, rehydrated, microwaved in 10 mM citrate buffer (pH 6.0) for 15 min, and incubated with mouse anti-BrdU antibody diluted 1:200 (BD Biosciences) and rabbit anti–phospho-histone H3 (pH3) antibody diluted 1:500 (marker for M-phase; Millipore) for 12 h at 4°C. Secondary antibodies were Alexa Fluor 488 conjugates diluted 1:1,000 (Invitrogen) or Cy3 conjugates diluted 1:500 (Jackson). The ratio of BrdU/pH3 double-positive cells to pH3-positive cells in the ventricular zone was calculated at 1, 1.5, and 2 h after a single injection of BrdU to determine the length of G2/M phase. A straight–line graph of the labeling index values (LIs)

at 1, 1.5, 2, 3.5, and 6.5 h allowed us to extrapolate to a y-axis intercept (the LIs at 0 h) and calculate the slope. Because the growth fraction (the ratio of proliferating cells) is nearly 1.0 in the ventricular zone of C57BL/6 mice, the LI at time = 0 and slope represent the ratio of S-phase to total cell cycle length (Ts/Tc) and the reciprocal of total cell cycle length (1/Tc), respectively. From these values (Ts/Tc and 1/Tc), Ts and Tc were calculated.

## In vivo labeling of annexin V

Mouse embryos at E15 were used for in vivo labeling of annexin V Alexa 488 conjugation (A13201; Thermo Fisher Scientific). Brain or other organs (heart, intestine) were directly incubated with 20 $\mu$l annexin V in 400 $\mu$l annexin V reaction buffer (25 mM Hepes, 140 mM NaCl, and 1 mM EDTA, pH 7.4) in 1.5-ml tubes for 15 min at room temperature. After three times wash by PBS, samples were fixed by 4% paraformaldehyde in 0.1 M phosphate buffer for 1 h at room temperature. Samples were further incubated with 20% and 30% sucrose in PBS for 12 h. 10 $\mu$m cryosections were prepared by Leica CM1850 (Leica). Sections were stained with DAPI (D523; DOJINDO) and coverslipped. Images were obtained by confocal microscopy (FV1200IXGP44; Olympus).

## 2D LC–MS/MS analysis

Second-passage neurospheres (48 h after plating) from four FTLD mouse models and littermate C57BL/6 mice were subjected to 2D LC MS/MS analysis. Adult mice at the ages of 1, 3, or 6 mo were euthanized and cerebral cortex was obtained. Neurospheres or mouse cerebral cortex samples were dissolved in lysis buffer (100 mM Tris–HCl [pH 7.5], 2% SDS, and 1 mM DTT) with protease inhibitor cocktail (#539134, 1:100 dilution; Calbiochem) for 1 h at 4°C. After centrifugation (12,000g for 10 min), supernatants (1.5 mg protein in 200 $\mu$l) were mixed with 100 $\mu$l of 1 M triethylammonium (TEAB) (pH 8.5), 3 $\mu$l of 10% SDS, and 30 $\mu$l of 50 mM tris-2-carboxyethylphosphine (TCEP). After incubation for 1 h at 60°C, cysteine residues were blocked with 15 $\mu$l of 200 mM methyl methanethiosulfonate (MMTS) for 10 min at room temperature. Samples were then digested with trypsin (150 $\mu$g) in 80 mM $CaCl_2$ for 24 h at 37°C. Phosphopeptides were enriched using the Titansphere Phos-Tio Kit (GL Sciences) and desalted using Sep-Pak Light C18 cartridge columns (Waters Corporation). The samples were then dried and dissolved with 25 $\mu$l of 100 mM TEAB (pH 8.5). The phosphopeptides were labeled separately using the iTRAQ Reagent multiplex assay kit (AB SCIEX Inc.) for 2 h at room temperature. After the samples were mixed together, the aliquots were dried and re-dissolved in 1 ml of 0.1% formic acid.

The labeled phosphopeptide samples were subjected to Strong Cation Exchange chromatography using TSK gel SP-2SW column (TOSOH) on a Prominence UFLC system. The flow rate was 1.0 ml/min of solution A (10 mM $KH_2PO_4$ [pH 3.0], 25% acetonitrile). Elution was performed with solution B (10 mM $KH_2PO_4$ [pH 3.0], 25% acetonitrile, and 1 M KCl) in a gradient ranging from 0 to 50%. The elution fractions were dried and dissolved in 100 $\mu$l of 0.1% formic acid.

Each fraction was analyzed using a DiNa Nano-Flow LC system (KYA Technologies Corporation) at a flow rate of 300 nl/min. For the Nano-LC, samples were loaded onto a 0.1 × 100 mm C18 column with solution C (2% acetonitrile and 0.1% formic acid) and eluted with a

gradient of 0–50% solution D (80% acetonitrile and 0.1% formic acid). The ion spray voltage to apply a sample from the Nano-LC to the Triple Time-of-Flight (TOF) 5600 System (AB SCIEX) was set at 2.3 kV. The information-dependent acquisition (IDA) setting was 400–1,250 m/z with two to five charges. The Analyst TF software (version 1.5; AB SCIEX) was used to identify each peptide. The quantification of each peptide was based on the TOF-MS electric current detected during the LC-separated peptide peak, adjusted to the charge/peptide ratio. The signals were analyzed by Analyst TF and processed using the ProteinPilot software (version 4; AB SCIEX) as described in the next section.

### Data analysis

Mass spectrum data of peptides were acquired and analyzed using Analyst TF. Using these results, we retrieved corresponding proteins from a public database of mouse protein sequences (UniProtKB/Swiss-Prot, downloaded from http://www.uniprot.org on June 22, 2010) using ProteinPilot (version 4; AB SCIEX), which employs the Paragon algorithm. Tolerance for the peptide search by ProteinPilot was set to 0.05 D for MS and 0.10 D for MS/MS analyses. "Phosphorylation emphasis" was set at the sample description, and "biological modifications" was set at the processing specification of Protein Pilot. The confidence score was used to evaluate the quality of identified peptides, and the deduced proteins were grouped using the Pro Group algorithm (AB SCIEX) to exclude redundancy. The threshold for protein detection was set at 95% confidence in ProteinPilot, and proteins with >95% confidence were accepted as identified proteins.

Quantification of peptides was performed through analysis of iTRAQ reporter groups in MS/MS spectra generated upon fragmentation in the mass spectrometer. For quantification of peptides, bias correction assuming that the total signal amount of each iTRAQ should be equal was used to normalize signals of different iTRAQ reporters. After bias correction, the ratio between reporter signals in VCP$^{T262A}$-KI mice and that of control mice (peptide ratio) was calculated.

These peptide ratios were imported to Excel files from summaries of ProteinPilot for further data analyses. Quantity of a phosphopeptide fragment was calculated as the geometric mean of signal intensities of multiple MS/MS fragments containing the phosphorylation site. Differences between the VCP$^{T262A}$-KI and control mice were statistically evaluated using Welch's test.

### Plasmid construction

To construct an expression vector of VCP-EGFP (pCI-VCP-EGFP), mouse VCP cDNA (nt 330–2,750 of NM_009503.4) was amplified from total RNA of mouse (C57BL/6) cerebral cortex using primers 5′-ACGTGTCGACACCATGGCTTCTGGAGCCGATTC-3′ and 5′-ACGTGGGC-CCCGCCATACAGGTCATCATCATT-3′, which contain ApaI and SalI sites, and subcloned into vector pEGFP-N1 (Clontech). To construct an expression vector of VCP-T262A-EGFP (pCI-VCP-T262A-EGFP), full-length VCP cDNA was amplified from total RNA of VCP$^{T262A}$-KI mice using primers 5′-ACGTGTCGACACCATGGCTTCTGGAGCCGATTC-3′ and 5′-ACGTGGGCCCCGCCATACAGGTCATCATCATT-3′, which contain ApaI and SalI sites, and subcloned into vector pEGFP-N1. To construct mouse MCM2-pCI-neo and mouse MCM3-pCI-neo, mouse MCM2 (nt 131–2,845 of NM_008564) and mouse MCM3 (nt 6–2,444 of NM_008563) were amplified from mouse MCM2-pYX-Asc plasmid (5696348;

DNAFORM) or mouse MCM3-pFLCI plasmid (I920096F09; DNAFORM) using the following primers: MCM2, 5′-TTTGAATTCACATGGCGGA-3′ and 5′-CTACAGCAGTTCTGAGTCGACTTT-3′; MCM3, 5′-ATGCCTCGAGGCATGG CGGGCACAGTAGTGC-3′ and 5′-ATGC GAATTCTCAGATAAGGAAGACGATGCC-3′. The primers contained EcoRI/SalI sites (MCM2) or XhoI/EcoRI sites (MCM3). Amplified inserts were subcloned into vector pCI-neo (Promega). Mutagenesis to introduce the S41A, S41D, and S41E mutations into MCM2 (MCM2-S41A-pCIneo, MCM2-S41D-pCIneo, and MCM2-S41E-pCIneo) was performed using the following primers: MCM2-S41A, 5′-ACCTCCGCCCCTGGCAGAGAC-3′ and 5′-GCCAGGGGCGG AGGTCAGGGCGTCA-3′; MCM2-S41D, 5′-TCCGACCCTGGCAGAGAC-3′ and 5′-AGGGTCGGAGGTCAGGGCGT-3′; MCM2-S41E, 5′-TCCGAACCTGGCAGAGA CCT-3′ and 5′-AGGTTCGGAGGTCAGGGCGT-3′. Mutagenesis to introduce the T719A, T719D, and T719E mutations into MCM3 (MCM3-T719A-pCIneo, MCM3-T719D-pCIneo, and MCM3-T719E-pCIneo) was performed using the following primers: MCM3-T719A, 5′-GTGCACGCAC CAAAGACTGACGATTCC-3′ and 5′-CTTTGGTGCGTGCACTTGAGGCATCTG-3′; MCM3-T719D, 5′-GTGCACGATCCAAAGACTGACGATTCC-3′ and 5′-CTTTGGA TCGTGCACTTGAGGCATCTG-3′; MCM3-T719E, 5′-GTGCACGAACCAAAGACTG ACGATTCC-3′ and 5′-CTTTGGTTCGTGCACTTGAGGCATCTG-3′.

### Micro-irradiation and time-lapse imaging

Laser micro-irradiation and signal acquisition from damaged areas were performed as described previously (Fujita et al, 2013). U2OS cells grown on 25-mm coverslips were treated with 2 $\mu$M Hoechst 33258 (H341; Dojindo) for 20 min to sensitize the cells to DSBs. Using the AIM4.2 software (Carl Zeiss) on an LSM510META microscope (Carl Zeiss), rectangle-shaped areas within cell nuclei were irradiated with a UV laser (maximum power: 30 mW, laser output: 75%, wavelength: 405 nm, iteration: five, pixel time: 12 $\mu$s: zoom 6), and time-lapse images were obtained every 30 s.

### Generation of gene networks based on protein–protein interactions (PPIs)

To generate a pathological gene network based on altered genes in VCP$^{T262A}$-KI mouse NSCs, we generated a list of genes encoding proteins whose phosphorylation was altered (Fig S7A). UniProt IDs were added to the genes on the list. Genes whose UniProt IDs were not listed in the genome network project (GNP) (https://cell-innovation.nig.ac.jp/GNP/index_e.html) database were removed from the list. The selected genes were used for the generation of a pathological PPI network based on the integrated database of GNP, including biomolecular interaction network database, the biological general repository for interaction datasets (BioGrid) (http://www.thebiogrid.org/), human protein reference database (http://www.hprd.org/), the IntAct molecular interaction database (IntAct) (http://www.ebi.ac.uk/intact/site/index.jsf), and molecular interactions database (MINT) (https://mint.bio.uniroma2.it/). One or two additional edges and nodes were added to selected nodes in the PPI database. A database of GNP-collected information was created on the Supercomputer System available at the Human Genome Center of the University of Tokyo.

### Generation of TDP$^{N267S}$-KI and CHMP2B$^{Q165X}$-KI mice

Generation of PGRN$^{R504X}$-KI mice was described previously (Fujita et al, 2018). In this study, heterozygous knock-in mice carrying the

CHMP2BQ165X mutation were generated in the C57BL/6J background. For construction of the targeting vector, a 5.7-kb *Cla*I–*Sal*I fragment amplified from a BAC (ID: RP23-13H5 or RP23-273E18) was subcloned into the *Cla*I–*Sal*I sites of pBS-DTA (Unitech), yielding Plasmid 1. Similarly, a 3.0-kb *Sac*II–*Not*I fragment was amplified from the same BAC clone and subcloned into pBS-LNL(+), which contains a Neo cassette (loxP-Neo-loxP) (Unitech), yielding Plasmid 2. Plasmid 2 was digested with *Sac*II–*Not*I, and a 4.6-kb fragment (1.6-kb Neo cassette + 3.0-kb fragment) was subcloned into the *Sac*II–*Cla*I sites of Plasmid 1; the resultant plasmid was used as the targeting vector. After linearization of targeting vector with *Sac*II, the fragment was electroporated into ES cells of the C57BL/6J background. Genotyping of ES clones was performed by PCR using following primers: 5′-ACGGAGTCTCTGCCTTACAAACTAC-3′ (forward) and 5′-CTTCCTCGTGCTTTACGGTATC-3′ (reverse), and positive clones were further confirmed by Southern blot analysis using 5′- and 3′-probes that had been amplified from genomic DNA with primers 5′-TAACGGTTCTATCTCAGGGTCAGTA-3′ and 5′-GTCACTTCTGTCTT-CACGGGTATGTG-3′ (5′ probe) or 5′-GTAGCCAGTCATACCAACATTGAC-3′ and 5′-TTGAGTAAGGTTTGAAGATCCAGAG-3′ (3′ probe). To confirm that the neomycin cassette was removed in F2 mice generated by crossing F1 and CAG-Cre mice, neomycin probe prepared by genomic PCR with primers 5′-GAACAAGATGGATTGCACGCAGGTTCTCCG-3′ and 5′-GTAGCCAACGCTATGTCCTGATAG-3′ was used for Southern blot analysis.

To construct the targeting vector for heterozygous knock-in of the TDP43N267S mutation, a 5.9-kb *Cla*I–*Sal*I fragment amplified from a BAC (ID: RP23-364M1 or RP23-331P21) was subcloned into the *Cla*I–*Sal*I sites of pBS-TK (Unitech), yielding Plasmid 1. Similarly, a 2.7-kb *Sac*II–*Not*I fragment was amplified from the same BAC clone and subcloned into pA-LNL(+), which contains a Neo cassette (loxP-Neo-loxP) (Unitech), yielding Plasmid 2. Plasmid 2 was digested with *Sac*II–*Not*I, and a 4.3-kb fragment (1.6-kb Neo cassette + 2.7-kb fragment) was subcloned into the *Sac*II–*Cla*I sites of Plasmid 1; the resultant plasmid was used as the targeting vector. After linearization of the targeting vector with *Sac*II, the fragment was electroporated into ES cells of the C57BL/6J background. Genotyping of ES clones was performed by PCR using the following primers: 5′-CGTGCAATCCATCTTGTTCAAT-3′ (forward) and 5′-CACCAGAATTA-GAACCACTGTAGGA-3′ (reverse), and positive clones were further confirmed by Southern blot analysis using 5′- and 3′′-probes amplified from genomic DNA with primers 5′-ATGCAGTTGATTA-CAGTCTTGCATA-3′ and 5′-GAGGATATTTGTAAGCAGGTTCACAC-3′ (5′ probe) or 5′-ATTGTCATGTATAAGTGCACCTGCT-3′ and 5′-GTCTT TGGCCTCTAAGTAGTGTCAT-3′ (3′ probe). To confirm that the neomycin cassette was removed in F2 mice generated by crossing F1 and CAG-Cre mice, neomycin probe amplified by genomic PCR with primers 5′-GAACAAGATGGATTGCACGCAGGTTCTCCG-3′ and 5′-GTAGCCAACGCTATGTCCTGATAG-3′ was used for Southern blot analysis.

CHMP2BQ165X-KI mice were discriminated from non-transgenic littermate by PCR using primers 5′-TGGATTTTATTTATGTCTGAATGTG-3′ and 5′-ATAAGCAACTTCACAAGGCATCTTA-3′, which amplify a DNA fragment containing the LoxP sequence. PCR conditions were as follows: 35 cycles of 94°C for 20 s for denaturation, 55°C for 40 s for annealing, and 72°C for 40 s for extension. A 267-bp fragment and a 374-bp fragment were amplified from background mice and

heterozygous CHMP2BQ165X-KI mice, respectively. TDP43N267S-KI mice were discriminated from non-transgenic littermates by PCR using primers 5′-ACAGTTGGGTGTGATAGCAGGTACT-3′ and 5′-TCGAGAATTA CAGGAATGTATCATC-3′, which amplify a DNA fragment containing the LoxP sequence. PCR conditions were as follows: 35 cycles of 94°C for 20 s for denaturation, 55°C for 30 s for annealing, and 72°C for 120 s for extension. A 240-bp fragment and a 347-bp fragment were amplified from background mice and heterozygous TDP43N267S-KI mice, respectively.

## Exon array

Total RNA samples were prepared from cerebral cortex of VCP-KI, PGRN-KI, CHMP2B-KI, TDP43-KI, and their background C57BL/6J mice by using RNeasy Mini kit (#74106; QIAGEN). Single-strand cDNA was made from 100 ng of total RNA with Ambion whole transcript (WT) expression kit for Affymetrix GeneChip WT expression arrays (Part Number 4425209 Rev.B). Terminal labeling of the cDNA probes and their hybridization to exon array were performed according to the manufacturer's instruction of Ambion WT expression kit (P/N 702808 Rev.1). The images were obtained by GeneChip Scanner 3,000 (#00-0074; Affymetrix) and analyzed with GeneChip Operating Software ver1.4 (#690036; Affymetrix) and Expression Console Software ver1.1 (#702387; Affymetrix). The probe sequence data were downloaded from the Affymetrix Web site (http://www.affymetrix.com/analysis/index.affx). Gene-level expression data were normalized using IterPLIER, which is an extension of PLIER algorithm (http://tools.thermofisher.com/content/sfs/brochures/exon_gene_signal_estimate_whitepaper.pdf). To determine the significance of the selected marker gene's expression change, *t* test was conducted under the assumption of equal variance in the signals between FTLD models and their background mice.

## Human patients

CSF samples were obtained from 94 patients (21 FTLD, 30 ALS, and 43 AD) diagnosed based on clinical symptoms, electrophysiological examination, and neuroimaging (magnetic resonance imaging (MRI), single-photon emission computed tomography, and positron emission tomography) at Nagoya University, Tohoku University, University of Tokyo, Choju Medical Institute of Fukushimura Hospital, and Higashi Matsudo Municipal Hospital. Normal control subjects were nine males (mean age, 73.7; range, 55–83 yr) and four females (mean age, 76; range, 71–79 yr). The MMSE scores of control subjects were greater than 23 (mean, 27.6; range, 24–30) when the CSF samples were collected. Neurological disease control subjects were eight males (mean age, 75; range, 60–83 yr) and six females (mean age, 64.5; range, 48–84 yr). The MMSE scores of neurological control subjects were greater than 21 (mean, 26.8; range, 22–30) when the CSF samples were collected. AD patients were 21 males (mean age, 75.9; range, 54–87) and 22 females (mean age, 71.1; range, 56–88 yr). Their MMSE scores were not greater than 26 (mean, 14.7; range, 0–26), and their mean FAB score was 9.3 (range, 6–13). FTLD patients were 10 males and 11 females with mean MMSE score of 17.4 (range, 7–30) and mean FAB of 8.6 (range, 3–16). ALS patients were 17 males, 12 females, and one person of unknown gender, with a mean ALSFRS-R score of 41.3 (range, 24–47). Twenty-nine ALS cases were diagnosed as "definite" and one as "possible." CSF samples from AD,

FTLD, and ALS patients were obtained using an approved protocol in accordance with the guidelines of the institutional review board of Nagoya University. CSF samples from control subjects were obtained using an approved protocol in accordance with the guidelines of the institutional review board of Tohoku University. Brain tissue samples from FTLD patients were collected using an approved protocol in accordance with the guidelines of the institutional review board of Choju Medical Institute of Fukushimura Hospital. Brain tissue samples from control patients were collected using an approved protocol in accordance with the guidelines of the institutional review board of Tokyo Medical and Dental University. The experiments were carried out in accordance with approved guidelines and regulations in line with the tenets of the Declaration of Helsinki. Informed consent was obtained from all patients.

## Sample preparation

Proteome analysis of CSF and serum was performed as described previously (Tagawa et al, 2015; Fujita et al, 2016). In brief, 50 $\mu$l of CSF from human patients was added to 50 $\mu$l of buffer containing 200 mM Tris–HCl (pH 7.5), 4% SDS, 2 mM DTT, 0.5 $\mu$l of Protease Inhibitor (Calbiochem), and 10 $\mu$l of 10× Phosphatase Inhibitor (Roche), and then incubated at 100°C for 15 min. For serum, 3 $\mu$l of serum was diluted with 47 $\mu$l distilled water and further diluted with 50 $\mu$l of buffer containing 200 mM Tris–HCl (pH 7.5), 4% SDS, 2 mM DTT, 0.5 $\mu$l of Protease Inhibitor, and 10 $\mu$l of 10× phosphatase inhibitor, and then incubated at 100°C for 15 min. The sample was mixed with 900 $\mu$l purified water and centrifuged at 16,000$g$ at 4°C for 10 min. The supernatant was passed through a 0.22-$\mu$m PVDF filter (Millipore) and 3k filter (Millipore). Aliquots (50 $\mu$l) were added to 25 $\mu$l of 1 M triethylammonium bicarbonate (TEAB) (pH 8.5), 0.75 $\mu$l of 10% SDS, and 7.5 $\mu$l of 50 mM tris-2-carboxyethyl phosphine, and then incubated for 1 h at 60°C. Cysteine residues were blocked for 10 min at 25°C with 10 mM methyl methanethiosulfonate. The samples were then digested with 1.5 $\mu$g (for CSF) and 22.5 $\mu$g (for serum) of trypsin (10:1 = protein:enzyme, w/w) in 24 mM CaCl$_2$ for 24 h at 37°C, desalted on a C18 spin column (MonoSpin C18; GL Sciences Inc.), dried, and dissolved in 35 $\mu$l of 0.1% formic acid.

## SWATH-mass analysis of CSF

Aliquots (30 $\mu$l) were applied to a C18 column (0.1 × 100 mm; KYA Technologies Corporation) with solution A (0.1% formic acid), eluted with a gradient of 2–41% solution A and B (99.9% acetonitrile/0.1% formic acid) at a flow rate of 300 nl/min using an Eksigent NanoLC-Ultra 1D Plus system (AB SCIEX), and then analyzed on a Triple TOF 5600 system (AB SCIEX) at an ion spray voltage of 2.3 kV. IDA was set at 400–1,000 $m/z$ with two to five charges, and the product ion MS/MS scan range was between 100 and 1,600 D, with an accumulation time of 100 ms for a spectral library. SWATH (sequential window acquisition of all theoretical mass spectra) acquisition was performed in 24 sequential windows of 25 D spanning from 400 to 1,000 D. SWATH acquisition of MS/MS spectral data was performed using the Analyst TF1.6 software for 100 ms per window, and the MS/MS spectral library was prepared using the Protein Pilot software (version 4.5). MS/MS spectral data were correlated with peptide data and LC retention time using the PeakView software (version

1.2.0.3; AB SCIEX). The MS/MS product ions from the same peptide were summed and used to indicate the quantity of the peptide.

## Mass data analysis

The raw MS/MS spectral data were processed, and the spectra of confidently identified species were extracted under the following conditions: extracted ion chromatogram (XIC) extraction window, 5 min; XIC width, 0.01 D. Extraction was performed using the SWATH 1.0 application in PeakView1.2. The XIC was displayed as a curve in a graph of LC retention time versus relative ion intensity in a small $m/z$ range. Fragment ion XICs were summed to obtain peptide peak areas, and areas for multiple peptides per protein were summed to obtain protein areas. Mass spectrum data of peptides were normalized against the total amount of all detected proteins or the total amount of peptides associated with MARCKS protein (referred to in the text as "whole proteins" or "total MARCKS," respectively) for each individual subject. To calculate sensitivity and specificity, phosphorylation levels greater than the mean + 2 SD of the control group were considered abnormal.

## AAV gene therapy of VCP^L198W iPSC–derived neurospheres and NSCs

Neurospheres were obtained as described above ("NSC differentiation from human iPSCs"). Neurospheres were seeded onto 6-cm dishes and infected the next day with AAV-[nes]-VCP-FLAG, AAV-[nes]-MCM3-T719A-FLAG, or AAV-[nes]-FLAG at a MOI of 5,000. Neurospheres were passaged twice every 7 d, and then dissociated in TrypLE Select containing 10 $\mu$M Y27632 (253-00513; Wako). Dissociated cells were re-seeded onto poly-L-ornithine (P3655; Sigma-Aldrich) and laminin (23016015; Gibco) in an eight-well chamber with DMEM/F12 (D6421; Sigma-Aldrich) supplemented with B27 (17504044; Thermo Fisher Scientific), GlutaMAX (35050061; Thermo Fisher Scientific) and penicillin/streptomycin (15140-122; Gibco). 5 d later, cells were fixed with 4% PFA and stained with anti-MAP2 antibody (T8660; Sigma-Aldrich) and DAPI.

## AAV gene therapy of model mice

AAV-pNestin-VCP-FLAG, AAV-pNestin-MCM3-T719A-FLAG, AAV-pNestin-FLAG, AAV-pCMV-VCP-FLAG, AAV-pCMV-MCM3-T719A-FLAG, and AAV-pCMV-FLAG were custom-ordered from VectorBuilder. For in utero gene therapy, female C57BL/6J mice were crossed with male VCP^T262A-KI mice, and AAV vectors (1.0 × 10^12 viral genomes [vg] in 100 $\mu$l) were injected into the tail vein of pregnant female mice. 1 or 6 mo after birth, male mice were euthanized, and brain weight was measured. Brain samples were subjected to histological analysis (HE staining, cortical layer marker staining, inclusion body staining for phosphorylated-TDP-43 and ubiquitin, and staining for DNA damage markers). At 6 mo of age, behavioral tests (Morris water maze test and Y-maze test) were performed. For adult gene therapy aimed at restoring adult neurogenesis, we injected AAV (1.0 × 10^12 vg in 2.5 $\mu$l) into the subarachnoid space (bregma +2.0 mm, lateral 0.6 mm right side) of male VCP^T262A-KI or male sibling non-transgenic mice at 19 wk, and euthanized them at 24 wk of age. BrdU

injection (50 µg/g body weight) was performed 1 wk after AAV injection (van Praag et al, 1999).

### In vitro phosphorylation

In vitro phosphorylation was induced by incubating 0.1 µg of MCM2 or MCM3 peptide (MCM2: SRRADALTSSPGRDLP, MCM3: ETQMPQVHTPKTDD; GenScript) with or without 3.7 µg of CDK1 (PV3292; Thermo Fisher Scientific), 5.8 µg CDK2 (C0495; Sigma-Aldrich), 5.8 µg of CDK4 (C31-10G; SignalChem Pharmaceuticals), 9.1 µg of DAPK1 (D01-11G; SignalChem Pharmaceuticals), 2.5 µg of ATM protein (14-933; Eurofins), 3.1 µg of DNA-PK (PV5866; Life Technologies), 8.1 µg ChK1 (14-346; Eurofins), or 6.1 µg ChK2 (14-347; Eurofins) in 30 ml of reaction buffer at 30°C for 30 min. Kinase levels were determined based on the molecular weight ratios between MCM substrates and kinases (substrates:kinases = 10:1). Reaction buffers for each kinase were prepared as follows: (CDK1, CDK2, CDK4, and DAPK1) 50 mM Hepes (pH 7.0), 5 mM $MnCl_2$, 10 mM $MgCl_2$, 1 mM DTT, 1 mM ATP; (ATM) 25 mM Hepes (pH 6.0), 1% glycerol, 1 mM ATP; (DNA-PK) 50 mM Hepes, 10 mM $MgCl_2$, 1 mM DTT, 1× activation buffer (manufacturer's product containing 2.5 µg/ml activator), and 1 mM ATP. The mixture was loaded onto a 0.1 × 100 mm C18 column with 2% acetonitrile and 0.1% formic acid solution. For LC, flow rate was 300 nl/min and ion spray voltage was 2.3 kV. The IDA setting was 400–1,250 m/z with two to five charges. The signals were analyzed by Analyst TF (version 1.5) and processed using the ProteinPilot software (version 4).

In vitro phosphorylation of TDP43 was induced by incubating 0.5 µg of human recombinant TDP43 (AP-190-100; R&D Systems) with 1.221 µg of human PKC-γ (P66-10G; SignalChem Pharmaceuticals) or 0.83 µg of human casein kinase 1 δ (C65-10G; SignalChem Pharmaceuticals) in 30 µl of reaction buffer at 30°C for 180 min. Reactants were mixed with an equal volume of sample buffer (62.5 mM Tris–HCl pH 6.8, 2% [wt/vol] SDS, 2.5% [vol/vol] 2-mercaptoethanol, 5% [vol/vol] glycerol, and 0.0025% [wt/vol] bromophenol blue) and subjected to SDS–PAGE.

### Behavioral test

Seven types of behavioral tests were performed for male mice, as described previously (Tagawa et al, 2015; Ito et al, 2015b) at the indicated ages. For the Morris water maze test, mice performed four trials (60 s) per day for 5 d, and the latency to reach the platform was measured. Also, on day 5, a probe test (in which the mouse's movement without the platform was traced for 60 s) was performed, and duration and time spent in the platform region were measured. The Y-shape maze consisted of three identical arms with equal angles between each arm (O'HARA & Co., Ltd). Mice were placed at the end of one arm and allowed to move freely through the maze during an 8-min session. The percentage of spontaneous alterations (indicated as an alteration score) was calculated by dividing the number of entries into a new arm that was different from the previous one by the total number of transfers from one arm to another. For the rotarod test, mice performed four trials (bar diameter: 3 cm, rod speed range: 3.5–35 rpm during trial) per day for 3 d, and the mean latency to falling off the rotarod was recorded. For the fear-conditioning test, the freezing response of mice was measured 24 h after the conditioning trial (65 dB white noise, 30 s + foot shock, 0.4 mA, 2 s) in the same chamber without a foot shock. For the open-field test, the duration of

the stay in the central area of an open-field space (50 × 50 × 40 cm [H]) was adopted as the index. In the light–dark box test, the duration of the stay in the light box (20 × 20 × 20 cm [H]) was measured. The elevated plus maze was set up 60 cm above the floor, and the duration of the stay in the open arms was measured.

### Two-photon microscopy

AAV1-EGFP with the synapsin I promoter (titer $1 × 10^{10}$ vg/ml, 1 µl) was injected into the retrosplenial cortex (anteroposterior, –2.0 mm from bregma and mediolateral 0.6 mm; depth, 1 mm) of 10-wk-old mice under anesthesia with 2.5% isoflurane. 2 wk later, a high-speed micro-drill was used to thin a circular area on the skull. The head of the animal was immobilized by attaching the head plate to a custom-machined stage mounted on the microscope table. Two-photon imaging was performed on a laser-scanning microscope system FV1000MPE2 (Olympus) equipped with an upright microscope (BX61WI; Olympus), a water-immersion objective lens (XLPlanN25xW; numerical aperture, 1.05), and a pulsed laser (Mai-TaiHP DeepSee, Spectra Physics). EGFP was excited at 890 nm and scanned at 500–550 nm. High-magnification imaging (101.28 × 101.28 µm; 1,024 × 1,024 pixels; 1-µm Z step) of cortical layer I was performed through the thinned-skull window at 5× digital zoom. For evaluation of a single cell volume, images were analyzed by Imaris x64 software (version 7.7.2; Bitplane).

### Ethics

This study was performed in strict accordance with the recommendations in the Guide for the Care and Use of Laboratory Animals of Japanese Government and the National Institutes of Health. All experiments were approved by the Committees on Gene Recombination Experiments, Human Ethics, and Animal Experiments of the Tokyo Medical and Dental University (G2018-082C, 2011-22-3/O2020-002, and A2019-218C2).

## Data Availability

All data generated or analyzed during this study are included in this article and its Supplemental Material files (Table S1).

## Supplementary Information

## Acknowledgements

This work was supported by the Strategic Research Program for Brain Sciences (SRPBS), Brain Mapping by Integrated Neurotechnologies for Disease Studies (Brain/MINDS), Acceleration Transformation Research for Medical Innovation (ACT-MS) from the Japan Agency for Medical research and Development (AMED); a Grant-in-Aid for Scientific Research on Innovative Areas (Foundation of Synapse and Neurocircuit Pathology, 22110001/

22110002) from the Ministry of Education, Culture, Sports, Science and Technology of Japan (MEXT); and a Grant-in-Aid for Scientific Research A (16H02655, 19H01042) from the Japan Society for the Promotion of Science (JSPS). We thank Dr. Minoru Nakayama and Ms. Ayumi Echigo (Toho University) for the *Drosophila* experiments, Ms. Tayoko Tajima (TMDU) for mouse and histology experiments, Dr. Xigui Chen (TMDU) for two-photon microscopy, and Dr. Emiko Yamanishi (TMDU) for iPS cell culture.

## Author Contributions

H Homma: data curation, formal analysis, and writing—original draft, review, and editing.
H Tanaka: data curation and writing—original draft, review, and editing.
M Jin: data curation and investigation.
X Jin: data curation and investigation.
Y Huang: data curation.
Y Yoshioka: data curation.
CJF Bertens: data curation.
K Tsumaki: data curation.
K Kondo: data curation.
H Shiwaku: data curation
K Tagawa: data curation.
H Akatsu: resources.
N Atsuta: resources.
M Katsuno: resources.
K Furukawa: resources.
A Ishiki: resources.
M Waragai: resources.
G Ohtomo: resources.
A Iwata: resources.
T Yokota: resources.
H Inoue: resources.
H Arai: resources.
G Sobue: resources.
M Sone: data curation and writing—original draft, review, and editing.
K Fujita: data curation and writing—original draft, review, and editing.
H Okazawa: conceptualization, supervision, funding acquisition, project administration, and writing—original draft, review, and editing.

## Conflict of Interest Statement

The authors declare that they have no conflict of interest.

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
