## [Reviewer comments · Life Science Alliance]

Life Science Alliance

DNA damage in embryonic neural stem cell determines FTLDs' fate via early-stage neuronal necrosis

Hidenori Homma, Hikari Tanaka, Meihua Jin, Xiaocen Jin, Yong Huang, Yuki Yoshioka, Christian Bertens, Kohei Tsumaki, Kanoh Kondo, Hiroki Shiwaku, Kazuhiko Tagawa, Hiroyasu Akatsu, Naoki Atsuta, Masahisa Katsuno, Katsutoshi Furukawa, Aiko Ishiki, Masaaki Waragai, Gaku Ohtomo, Atsushi Iwata, Takanori Yokota, Haruhisa Inoue, Hiroyuki Arai, Gen Sobue, Masaki Sone, Kyota Fujita, and Hitoshi Okazawa

DOI: <https://doi.org/10.26508/lsa.202101022>

Corresponding author(s): Hitoshi Okazawa, tokyo medical and dental university

Review Timeline:	Submission Date:	2021-01-20
	Editorial Decision:	2021-04-07
	Revision Received:	2021-05-07
	Editorial Decision:	2021-05-21
	Revision Received:	2021-05-24
	Accepted:	2021-05-24

Scientific Editor: Shachi Bhatt

Transaction Report:

April 7, 2021

Re: Life Science Alliance manuscript #LSA-2021-01022-T

Prof. Hitoshi Okazawa
tokyo medical and dental university
1-5-45, Yushima, Bunkyo-ku
Tokyo 113-8510
Japan

Dear Dr. Okazawa,

Thank you for submitting your manuscript entitled "DNA damage in embryonic neural stem cell determines FTLDs' fate via early-stage neuronal necrosis" to Life Science Alliance (LSA). The manuscript was assessed by expert reviewers, whose comments are appended to this letter. We would like to invite you to submit a revised version of this manuscript back to LSA that addresses all of the reviewers' points.

We apologize for this unusual and extended delay in getting back to you, caused due to the difficulty in finding reviewers for this study. Ultimately, we were able to secure 2 experts to review the study. As you will note from the comments below, the referees appreciate the extensive amount of data included in the manuscript but do think that additional analyses are required to substantiate some of these claims and the revised manuscript needs to clear up the presentation. We would also encourage you to substantially edit the manuscript text and the figures.

Thank you for this interesting contribution to Life Science Alliance. We are looking forward to receiving your revised manuscript.

Sincerely,

Shachi Bhatt, Ph.D.
Executive Editor
Life Science Alliance
<http://www.lsajournal.org>
Tweet @SciBhatt @LSAJournal

- A letter addressing the reviewers' comments point by point.
- An editable version of the final text (.DOC or .DOCX) is needed for copyediting (no PDFs).
- High-resolution figure, supplementary figure and video files uploaded as individual files: See our detailed guidelines for preparing your production-ready images, <https://www.life-science-alliance.org/authors>
- Summary blurb (enter in submission system): A short text summarizing in a single sentence the study (max. 200 characters including spaces). This text is used in conjunction with the titles of papers, hence should be informative and complementary to the title and running title. It should describe the context and significance of the findings for a general readership; it should be written in the present tense and refer to the work in the third person. Author names should not be mentioned.

B. MANUSCRIPT ORGANIZATION AND FORMATTING:

Reviewer #1 (Comments to the Authors (Required)):

The authors have created a mouse model for FTL, and identified a fundamental understanding of aberrant DNA damage repair and neurogenesis in the model. They have also compared their findings with other models and ultimately confirmed their markers in human patient samples. Overall the study is extremely comprehensive and highly detailed. The results of studies from human

samples could be further detailed.

There does seem to be a lot of information for 1 paper. Some of my comments are as follows

Line 151-152 - Please describe the phenotype that you mention here

Fig 1C, n=30, is that a typo?

Supplemental fig S2 a - TDP43 staining appears more diffuse than aggregate, this could be due to lower resolution. Kindly address.

Line 181 - revealed or showed, not both

Fig 1D, at E18 and 6M, from the image it appears that the loading in WT is higher, indicating higher actual levels of VCP, if that is not true, a better representative image would increase the quality of that figure.

Fig 3c - Western blot shows Chk2 phosphorylation, kindly explain line 302.

Overall, the identification of pSer46-MARCKS in CSF as a biomarker has a practical application. However, the fact that gene therapy in utero and not as an adult rescued the FTLD phenotype raises the concerns of the applicability of detection and treatment currently. Could the authors address if and how this finding might have a therapeutic potential?

Reviewer #2 (Comments to the Authors (Required)):

In their manuscript "DNA damage in embryonic neural stem cell determines FTLDs' fate via early-stage neuronal necrosis" Homma et al link DNA damage during embryonic/early postnatal neurogenesis to a FTLD phenotype in a mouse model of FTLD. The findings are interesting, albeit in the context of FTLD models not entirely novel (previous work needs to be cited; e.g., their statement "it is not clear whether or how DNA repair influences FTLD Pathology" is not entirely true given for example Lopez-Gonzalez et al 2019 PNAS <https://www.pnas.org/content/116/19/9628>). The authors also need to discuss the relatively large amount of links between DNA damage and altered repair in FUS and C9orf72 mutants (associated with ALS but also FTD, a subgroup of FTLDs).

The study represents a tour de force with a large amount of data. The authors come to 3 main conclusions.

i) "In VCPT262A-KI mice, DNA repair in embryonic NSCs led to accumulation of DNA damage, DDR activation, MCM3 phosphorylation, partial G1/S arrest, excessive neurogenesis, depletion of the NSC pool, abnormal layer structure, and sustained DNA damage in neurons linked to early-stage neuronal necrosis."

The authors need to show in their main figures the relevant data (that are now mixed between main and supplement). What is really causing the (unusual) microcephalic phenotype that is "lost" with advancing age? Reduced proliferation (does not seem to be the case, S4A), enhanced neuronal differentiation (remains unclear as proper analyses of embryonic neuronal differentiation is missing), enhanced cell death (should be properly quantified)? The authors really need to focus on their main findings in the main figures. In its current form it is not clear how they come to the conclusion on page 7: "Collectively, these findings indicate that elevated neurogenesis and depletion of stem cell pool was the sole reason for microcephaly."

Furthermore, the authors do not discuss how microcephaly is "rescued" at 6 months of age. How is that happening? What is the mechanism? The analyses of different neuronal layers in Figure 1 is

not conclusive in that regard.

ii) "Our second major finding is the discovery that DNA damage drives early-stage neuronal necrosis in FTLD."

Again, the authors should really focus on their main findings (e.g., the proteomics data are interesting for MCM3 discovery but take too much space in main figure). The key findings are: in different models there are signs of elevated TRIAD necrosis. That has to be shown in main but not "diluted" with controls, different species, in vitro data. Furthermore, the authors need to relate their findings to previous work linking other FTLD mutants to DNA damage (see above).

iii) "The third major finding from this study is the discovery that multiple types of FTLD share a pathological cascade from impairment of DNA repair in NSCs to aggregation pathology in adulthood, mediated by early-stage neuronal necrosis."

It remains somewhat unclear (and not properly tested, e.g., Figure 1h misses cellular phenotyping) if DNA damage is indeed carried over from NSCs. This needs to be addressed. The idea that impaired DNA repair may represent a common feature is interesting and some data are suggestive. However, as outlined above, it remains unclear if NSC DNA damage is truly causative.

Overall, the authors here present an amazing amount of data. That is great. However, they fail to clearly present their key findings and the core messages are diluted by excessive inclusion of data into main figures that are interesting but of minor relevance. The authors should very substantially edit their main figures (in its current format they are also very hard to read and some panels are much too small).

We suggest that Figure 1 shows the developmental and adult brain phenotype of KI mice (in vivo, with proper analyses of cellular phenotypes and not just brain weight), Figure 2 shows discovery of MCM3 alterations, Figure 3 shows rescue with expression of MCM3, Figure 4 shows relevance of pSer46-MARCKS activation of in other models of FTLD and human tissues, Figure 5 shows elevated levels of pSer46-MARCKS in CSF of patients.

If the authors can convincingly present their core data (that are partially hidden in the supplement) the study will be of interest to the field.

Reviewer #1 (Comments to the Authors (Required)):

The authors have created a mouse model for FTLN, and identified a fundamental understanding of aberrant DNA damage repair and neurogenesis in the model. They have also compared their findings with other models and ultimately confirmed their markers in human patient samples. Overall the study is extremely comprehensive and highly detailed. The results of studies from human samples could be further detailed.

There does seem to be a lot of information for 1 paper.

>>> We really thank reviewer 1 for the kind evaluation of our paper and various advices to improve our manuscript.

Some of my comments are as follows

Line 151-152 - Please describe the phenotype that you mention here

>>> Here, we mean external appearance and body weight gain of two lines. We added the information in the text and in Supplementary Figure S1D.

Fig 1C, n=30, is that a typo?

>>> We observed three mice (N=3) respectively in C57BL/6J and VCP-KI. With each mouse, we made 10 tissue sections (slides) from Bregma -1.7mm, and measured cortex thickness at 1.0mm lateral from midline. We used n=30 (10 slides x 3 mice = 30 slides in total) for the total slide number in a genotype group.

Supplemental fig S2 a - TDP43 staining appears more diffuse than aggregate, this could be due to lower resolution. Kindly address.

>>> We replaced the image with a better one in Supplementary Fig S2A.

Line 181 - revealed or showed, not both

>>> We corrected the error.

Fig 1D, at E18 and 6M, from the image it appears that the loading in WT is higher, indicating higher actual levels of VCP, if that is not true, a better representative image would increase the quality of that figure.

>>> We agree with the reviewer that it is not so representative. We replaced it with a better image following the kind advice.

Meanwhile, we believe the reviewer knows there are always variation in each sample and in each blot in this technique, as shown as variations in the right graph. Therefore, completely equal blot, if it occurs in other papers, is impressive but somewhat doubtful in my eye.

Of course, we appreciate very much that the reviewer kindly advised us to exclude unnecessary questions from readers.

Fig 3c - Western blot shows Chk2 phosphorylation, kindly explain line 302.

>>> We had made a mistake in the text. We corrected it and now described that Chk2 is activated.

Overall, the identification of pSer46-MARCKS in CSF as a biomarker has a practical application. However, the fact that gene therapy in utero and not as an adult rescued the FTLD phenotype raises the concerns of the applicability of detection and treatment currently. Could the authors address if and how this finding might have a therapeutic potential?

>>> We thank the reviewer for the critical comments. We added a paragraph to Discussion about the concerns of gene therapy in utero.

Reviewer #2 (Comments to the Authors (Required)):

In their manuscript "DNA damage in embryonic neural stem cell determines FTLDs' fate via early-stage neuronal necrosis" Homma et al link DNA damage during embryonic/early postnatal neurogenesis to a FTLD phenotype in a mouse model of FTLD. The findings are interesting, albeit in the context of FTLD models

not entirely novel (previous work needs to be cited; e.g., their statement "it is not clear whether or how DNA repair influences FTLD Pathology" is not entirely true given for example Lopez-Gonzalez et al 2019 PNAS <https://www.pnas.org/content/116/19/9628>). The authors also need to discuss the relatively large amount of links between DNA damage and altered repair in FUS and C9orf72 mutants (associated with ALS but also FTD, a subgroup of FTLDs).

>>> We appreciate very much the great effort and the critical comment from reviewer 2. In Introduction, we added a new paragraph on DNA damage repair, DNA damage and DNA damage response signaling in Introduction with referring the critical paper on Ku80 (Lopez-Gonzalez et al 2019 PNAS) as suggested by reviewer 2.

The study represents a tour de force with a large amount of data. The authors come to 3 main conclusions.

i) "In VCPT262A-KI mice, DNA repair in embryonic NSCs led to accumulation of DNA damage, DDR activation, MCM3 phosphorylation, partial G1/S arrest, excessive neurogenesis, depletion of the NSC pool, abnormal layer structure, and sustained DNA damage in neurons linked to early-stage neuronal necrosis."

The authors need to show in their main figures the relevant data (that are now mixed between main and supplement).

>>> We restructured main and supplementary figures by exchanging panels to focus on our claims with main figures, by deleting redundant or unessential figures including data of Drosophila (old Fig 2a, 2c, 3f, 3g, 3h, 3i, Sup Fig S1d, S5, S14), by adding new data requested by reviewers (new Fig. 1J, 1K), and by adding appropriate cellular markers (new Fig 2E, Sup Fig S13B).

Reduced proliferation (does not seem to be the case, S4A),

>>> Old/new Supplementary Fig S4A show pH3-positive cells that are in M-phase. As shown in Supplementary Fig S5, and as summarized in S5E, cell proliferation reflects all cell cycle phases, and in the case of VCP mutation G1/S

times are elongated to reduce proliferation. M-phase time is not remarkably changed consistently with that pH3-positive cells are not remarkably changed.

enhanced neuronal differentiation (remains unclear as proper analyses of embryonic neuronal differentiation is missing),

>>> We calculated the ratio of BrdU-positive neurons (BrdU+/MAP2+) to total BrdU-positive cells (BrdU+) reflecting neurogenesis ratio, and confirmed enhanced neuronal differentiation (new Supplementary Fig S6B).

enhanced cell death (should be properly quantified)?

>>> We performed in vivo annexin V staining and showed no change in cell death during neurogenesis from NSCs to neurons (new Supplementary Fig S6D).

The authors really need to focus on their main findings in the main figures. In its current form it is not clear how they come to the conclusion on page 7:

"Collectively, these findings indicate that elevated neurogenesis and depletion of stem cell pool was the sole reason for microcephaly."

>>> We restructured main figures as described above, and revealed the straight line to the conclusion as described in the following answers.

What is really causing the (unusual) microcephalic phenotype that is "lost" with advancing age?

>>> We moved Golgi staining image to Fig 1, which show hyper-arborization of cortical neurons (new Fig. 1G, H, I), and added evaluation of single neuron volume by two-photon microscopy (new Fig. 1J). We also added evaluation of glia numbers per cortex tissue volume (new Fig. 1K). These data explain the recovery mechanism from microcephaly during aging.

Furthermore, the authors do not discuss how microcephaly is "rescued" at 6 months of age. How is that happening? What is the mechanism?

>>> We added the experiments to indicate increase of single neuron volume and increase of glia number underlie the recovery (new Fig. 1G, H, I, J, K).

The analyses of different neuronal layers in Figure 1 is not conclusive in that regard.

>>> We agree the change of cortical layers do not explain the recovery.

ii) "Our second major finding is the discovery that DNA damage drives early-stage neuronal necrosis in FTLD."

Again, the authors should really focus on their main findings (e.g., the proteomics data are interesting for MCM3 discovery but take too much space in main figure).

>>> We deleted old Fig 2a and 2c.

The key findings are: in different models there are signs of elevated TRIAD necrosis. That has to be shown in main but not "diluted" with controls, different species, in vitro data.

>>> We deleted such diluting controls (old Fig 6c, old Sup Fig S14), different species (*Drosophila* data in old Sup Fig S5), and in vitro data (old Fig 3f, 3g, 3h, 3i).

Furthermore, the authors need to relate their findings to previous work linking other FTLD mutants to DNA damage (see above).

>>> We added some consideration to DNA damage and neurodegeneration including the case of FTLDs in the first paragraph of Discussion.

iii) "The third major finding from this study is the discovery that multiple types of FTLD share a pathological cascade from impairment of DNA repair in NSCs to aggregation pathology in adulthood, mediated by early-stage neuronal necrosis."

It remains somewhat unclear (and not properly tested, e.g., Figure 1h misses cellular phenotyping) if DNA damage is indeed carried over from NSCs. This needs to be addressed.

>>> We added data of co-staining with γ H2AX and MAP2/Sox2 in new Figure 2B (=old Figure 1h), which showed neurons differentiated from NSCs keep DNA damage during embryo. In addition, we showed γ H2AX/MAP2 and 53BP1/MAP2 co-staining in new Figure 2E, which indicated DNA damage remaining in neurons in adulthood.

The idea that impaired DNA repair may represent a common feature is interesting and some data are suggestive. However, as outlined above, it remains unclear if NSC DNA damage is truly causative.

>>> We appreciate kind understanding of the reviewer. Yes, we identified phosphorylation of MCM3 is a trigger of DNA damage in NSCs. Commitment to MCM3 during embryo but not adulthood was successful to recover adult phenotype of four FTLN models in pathology and behavior. Therefore, we think this is a strong support. Meanwhile, as the reviewer 2 may consider, we do not intend to exclude other pathologies that initiate from later time points. Addition, combination and integration of multiple pathological elements would lead to final neurodegeneration.

Overall, the authors here present an amazing amount of data. That is great. However, they fail to clearly present their key findings and the core messages are diluted by excessive inclusion of data into main figures that are interesting but of minor relevance.

The authors should very substantially edit their main figures (in its current format they are also very hard to read and some panels are much too small).

>>> We followed the advice of reviewer 2 to restructure the line of main figures, as described below.

We suggest that Figure 1 shows the developmental and adult brain phenotype of KI mice (in vivo, with proper analyses of cellular phenotypes and not just brain weight), Figure 2 shows discovery of MCM3 alterations, Figure 3 shows rescue with expression of MCM3, Figure 4 shows relevance of pSer46-MARCKS activation of in other models of FTLN and human tissues, Figure 5 shows elevated levels of pSer46-MARCKS in CSF of patients.

If the authors can convincingly present their core data (that are partially hidden in the supplement) the study will be of interest to the field.

>>> We faithfully followed the advice from Figure 1 to Figure 5. Meanwhile we consider insertion of a figure on DNA damage and another figure on relevance of pSer46-MARCKS activation in VCP-FTLD model would make the flow smoother. It is also because that LSA accepts a higher number of figures usually.

Therefore, we consulted with the editor about the number, contents and order of main figures, and received his approval. We hope the reviewers also kindly accept this line of main figures.

May 21, 2021

RE: Life Science Alliance Manuscript #LSA-2021-01022-TR

Prof. Hitoshi Okazawa
tokyo medical and dental university
1-5-45, Yushima, Bunkyo-ku
Tokyo 113-8510
Japan

Dear Dr. Okazawa,

Thank you for submitting your revised manuscript entitled "DNA damage in embryonic neural stem cell determines FTLDs' fate via early-stage neuronal necrosis". The revision has now been reviewed by the original referees, and their reports are appended at the end of this email.

As you will see from the reviewers' comments below, the reviewers are happy with the revisions, but do think that some minor data presentation changes still need to be incorporated, and text changes included so as to avoid over-interpretations. We would be happy to publish your paper in Life Science Alliance pending these minor revisions raised by the reviewers and final revisions necessary to meet our formatting guidelines.

Along with the reviewers' points, we also request you to address the following formatting concerns:

- there is a name discrepancy between how the name of one of your co-authors is presented. Please correct: Gaku Otomo in ms file vs. Gaku Ohtomo in the system
- please add ORCID ID for the corresponding author-you should have received instructions on how to do so
- please add a conflict of interest statement to your main manuscript text
- please add callouts for Figure S10A-D to your main manuscript text
- please add your table legends to the main manuscript text after the main and supplementary legends
- please provide higher quality images for blots shown in Figure 2
- please add missing scale bars for Figure 4E, 5B, 6B

A. FINAL FILES:

B. MANUSCRIPT ORGANIZATION AND FORMATTING:

Sincerely,

Shachi Bhatt, Ph.D.
Executive Editor
Life Science Alliance
<http://www.lsjournal.org>
Tweet @SciBhatt @LSAJournal

Reviewer #1 (Comments to the Authors (Required)):

Upon reviewing the revisions submitted by the authors, it appears they have addressed most of my concerns. A couple of points

Fig 1c - Even though there are 10 slides, the number of biological replicates is 3 therefore the n should be 3, in the methods section, the authors can mention the details of using 10 sections (technical replicates) per sample. Please change it to n=3

Fig 1D - Yes the image is better now. While I am quite aware of the challenges of completely equal loading and expression, a figure is called 'representative' so it best represents the analysis. Thank you for addressing this.

Thank you!

Reviewer #2 (Comments to the Authors (Required)):

The authors have now submitted a restructured, revised manuscript. They were able to clarify their main findings and previous concerns were sufficiently addressed. However, the authors need to tone down several claims as their data do not finally prove some of the claims.

Examples:

First sentence discussion:

"is the discovery that developmental stress due to DDR triggered by accumulation of unrepaired DNA damage in NSCs is the ultra-early stage pathology responsible for initiating FTLD"

The data presented here are suggestive. But not conclusively proving that this is the main/only cause of FTLD pathology. The last sentence of the discussion is much more appropriate.

Instead of discussing extensively very generic aspects of gene therapy (which are not very "exciting") the authors should rather discuss their new findings that neurons enlarge in the mutant (Figure 1J-K). This is not discussed at all. How does that finding relate to all other data? Furthermore, the rescue of microcephaly needs much more attention in the discussion.

The figures should be still improved and "clarified" (e.g., a quarter for Figure 4 is now "abbreviation..." - that is not perfect figure design). All shadings in the background (e.g., Figure 3 and 7) should be removed.

The authors present interesting data and a large amount of work. The presentation of the findings improved. However, the authors should still try to improve the presentation of their work and carefully edit the manuscript.

Reviewer #1 (Comments to the Authors (Required)):

Upon reviewing the revisions submitted by the authors, it appears they have addressed most of my concerns. A couple of points

Fig 1c - Even though there are 10 slides, the number of biological replicates is 3 therefore the n should be 3, in the methods section, the authors can mention the details of using 10 sections (technical replicates) per sample. Please change it to n=3

>>> We corrected the corresponding parts in Figure legend and Figure of Fig 1c.

Fig 1D - Yes the image is better now. While I am quite aware of the challenges of completely equal loading and expression, a figure is called 'representative' so it best represents the analysis. Thank you for addressing this.

>>> Thank you very much for your kind understanding.

Thank you!

Reviewer #2 (Comments to the Authors (Required)):

The authors have now submitted a restructured, revised manuscript. They were able to clarify their main findings and previous concerns were sufficiently addressed. However, the authors need to tone down several claims as their data do not finally prove some of the claims.

Examples: First sentence discussion:

"is the discovery that developmental stress due to DDR triggered by accumulation of unrepaired DNA damage in NSCs is the ultra-early stage pathology responsible for initiating FTLD" The data presented here are suggestive. But not conclusively proving that this is the main/only cause of FTLD pathology. The last sentence of the discussion is much more appropriate.

>>> We agree that our claim is not completely proven like in the case of proofs in the fields of mathematics or physics, though we think that our new pathology is at least one of the major factors especially in the earliest phase of FTLDs' pathologies, given that AAV-mediated intervention at embryonic stage had a great impact on the adulthood pathology.

Following the advice of the reviewer, we changed the expression of the first sentence in Discussion.

Instead of discussing extensively very generic aspects of gene therapy (which are not very "exciting") the authors should rather discuss their new findings that neurons enlarge in the mutant (Figure 1J-K). This is not discussed at all. How does that finding relate to all other data? Furthermore, the rescue of microcephaly needs much more attention in the discussion.

>>> We appreciate the comment from the reviewer. Microcephaly and rescue, which might be related to neuronal size, are important findings from this study and we had not discussed enough. We added one paragraph in Discussion for this issue.

The figures should be still improved and "clarified" (e.g., a quarter for Figure 4 is now "abbreviation..." - that is not perfect figure design). All shadings in the background (e.g., Figure 3 and 7) should be removed.

>>> We corrected Figure 4, 3 and 7.

The authors present interesting data and a large amount of work. The presentation of the findings improved. However, the authors should still try to improve the presentation of their work and carefully edit the manuscript.

>>> We hope abovementioned corrections are sufficient for final decision.

May 24, 2021

RE: Life Science Alliance Manuscript #LSA-2021-01022-TRR

Prof. Hitoshi Okazawa
tokyo medical and dental university
1-5-45, Yushima, Bunkyo-ku
Tokyo 113-8510
Japan

Dear Dr. Okazawa,

Thank you for submitting your Research Article entitled "DNA damage in embryonic neural stem cell determines FTLDs' fate via early-stage neuronal necrosis". It is a pleasure to let you know that your manuscript is now accepted for publication in Life Science Alliance. Congratulations on this interesting work.

DISTRIBUTION OF MATERIALS:

Again, congratulations on a very nice paper. I hope you found the review process to be constructive and are pleased with how the manuscript was handled editorially. We look forward to future exciting submissions from your lab.

Sincerely,

Shachi Bhatt, Ph.D.

Executive Editor

Life Science Alliance

<http://www.lsjournal.org>
